# Ovalbumin-specific regulatory T cells differentiated from the naïve phenotype (CD44$^{lo}$CD62L$^{hi}$) in mesenteric lymph nodes stably suppress enteropathy even in severe food-allergic mice

Kyoko Shibahara[1☯], Tomohiro Hoshino[1☯], Haruka Nakanishi[1], Kosuke Nishitsuji[1], Kohei Soga[1], Yoshiyo Bamba[1], Satoshi Hachimura[1], Haruyo Nakajima-Adachi[1,2]*

1 Research Center for Food Safety, Graduate School of Agricultural and Life Sciences, The University of Tokyo, Bunkyo-ku, Tokyo, Japan, 2 Department of Immunobiology and Biofunctional Research, Graduate School of Agricultural and Life Sciences, The University of Tokyo, Bunkyo-ku, Tokyo, Japan

☯ These authors contributed equally to this work.
* haruyona@g.ecc.u-tokyo.ac.jp

## Abstract

Impaired expansion, stability, and function of regulatory T cells (Tregs) are reported in patients with severe allergy. Transfer of Tregs is a potential means of treating severe food allergy; however, methods to obtain allergen-specific Tregs with stable regulatory activities are needed. To achieve our goal, we examined the characteristics of allergen-specific Tregs by comparing two mouse strains transgenic for the ovalbumin (OVA)-specific T cell receptor gene: Rag23−3 and RagD10 mice (OVA23−3 and DO11.10 crossed with *Rag2* knockout mice, respectively). RagD10 is a tolerant model, whereas Rag23−3 shows severe allergy when fed egg white (EW). To examine the differentiation of CD4$^+$ T cells into Foxp3$^+$ Tregs (induced Tregs; iTregs), CD4$^+$ T cells or whole cells from mesenteric lymph nodes or spleens were cultured under Treg-polarization conditions and stimulated with either a combination of anti-CD3 and anti-CD28 antibodies or OVA plus antigen-presenting cells. After stimulation with the antibodies, iTregs were induced at comparable levels from CD4$^+$ T cells from untreated Rag23−3 and RagD10 mice. Transfer of the resultant iTregs from untreated Rag23−3 mice suppressed allergic responses in EW-fed Rag23−3 mice. In contrast, stimulation with OVA plus antigen-presenting cells prevented the differentiation of iTregs from CD4$^+$ T cells from untreated Rag23−3 mice, suggesting that OVA-induced T-cell receptor signaling inhibits effective Treg differentiation. Furthermore, antibody-mediated differentiation afforded significantly more iTregs differentiation of naïve (CD44$^{lo}$CD62L$^{hi}$) CD4$^+$ T cells than of effector/effector memory (CD44$^{hi}$CD62L$^{lo}$) T cells isolated from the mesenteric lymph nodes of EW-fed Rag-23–3 mice. Excessive production of interleukin-4 and interferon-gamma by CD4$^+$ T cells from EW-fed

**Data availability statement:** All Excel files containing the data underlying the results presented in our study (Fig 1~Fig 5 and S1 Fig ~ S5 Fig) are available at the URL address of UTokyo Repository (http://hdl.handle.net/2261/0002013444).

**Funding:** This work was supported by grants from the Kieikai Research Foundation (HNA, Grant number; 2017S063, https://www.nakashima-foundation.org/kieikai/) and Grant-in-Aid for Scientific Research (B) from the Japan Society for the Promotion of Science (JSPS)(SH, Grant number; 26292065, https://www.jsps.go.jp/j-grantsinaid/index.html). The analysis for Fig. 5 and S5 Fig was funded by Meiji Holdings Co., Ltd (HNA and SH). Meiji Holdings Co., Ltd., we have no grant number assigned to this program grant in accordance with the nature of the grant, because The University of Tokyo's Corporate Sponsored Research Programs are programs established to conduct research on common issues that are of highly public nature in collaboration with the University of Tokyo, using funds received from the private sector and other external organizations. [Corporate Sponsored Research Programs | The University of Tokyo (u-tokyo.ac.jp)].The funders had no role in study design, data collection and analysis, decision to publish, or preparation of the manuscript.

**Competing interests:** We have read the journal's policy and the authors of this manuscript have the following competing interests: [Meiji Holdings Co.,Ltd.]. This does not alter our adherence to PLOS ONE policies on sharing data and materials.

Rag23−3 mice significantly inhibited Treg induction in RagD10 mice, suggesting the severe allergic cytokine milieu likely prevents their differentiation. However, our study showed that allergen-specific Tregs with regulatory activity can be obtained from naïve CD4+ T cells from the intestinal immune system of mice even with severe allergy.

## Introduction

Over the last two decades, the prevalence of food allergies has been increasing in developing countries; however, a standard treatment is yet to be established [1,2]. Oral immunotherapy, which involves feeding an allergic individual an increasing amount of allergen with the goal of increasing the threshold that triggers a reaction, has been shown to confer food-allergic patients with tolerance to allergens, but there are many issues related to safety (e.g., severe anaphylaxis) and efficacy that must be overcome before this approach can be more widespread use [3–9].

At the cellular level, allergen-specific regulatory T cells (Tregs) are considered to be key players in inhibiting allergic responses to food and maintaining tolerance [10–12]. Indeed, several studies have shown that individuals with food allergies who were able to tolerate by oral immunotherapy, or children who had outgrown a food allergy, possess allergen-specific Tregs with stronger abilities to inhibit the development of allergic inflammation compared with those in control patients still with severe food allergy [3,12,13]. This suggests that induction of allergen-specific Tregs in individuals with food allergies may confer stable tolerance in these individuals. Thus, just as in other inflammatory disease [14,15], Treg-transfer may be a promising means of treating food allergy.

To understand more about the characteristics and roles of allergen-specific Tregs, methods to induce their differentiation and to analyze their roles in food-allergic patients are desired; however, many allergists are trying to find markers for stable Tregs specifically activated by antigens in peripheral blood, but the goal has not been achieved, although CD137 has been proposed as a candidate marker [16–18]. In addition, Tregs induced under allergic inflammation conditions are highly plastic and easily induced to change into effector T cells, making their identification more difficult [16–18]. Indeed, the differentiation of antigen-specific Tregs occurs continuously in the intestine [19], and it is hypothesized that food allergy develops as a result of shifts in the balance of the activities of these antigen-specific Tregs and effector T cells [11]. This hypothesis is supported by reports showing that components of the microbiome destabilized Tregs in the small intestine, causing them to be easily induced to change to effector T cells [20,21]. To achieve the goal of adoptive transfer of allergen-specific Tregs for the treatment of food allergies, a more detailed understanding of how allergen-specific Tregs differ between individuals with and without allergy and in the various tissues is needed.

Here, to examine the differences between allergen-specific Tregs in healthy and allergic individuals, we used two mouse models of human food-allergic symptoms

(Rag23−3 and RagD10), as reported previously [22,23]. Briefly, the two models are both transgenic for the ovalbumin (OVA)-specific T-cell-receptor gene and lack recombination-activating gene 2 (*Rag2*). As such, these mice produce only OVA-specific T cells in response to OVA challenge, which allows for the study of the functions of T cells without the need to consider B cell responses. Rag23−3 mice are a cross between OVA23−3 mice and *Rag2* knockout mice and are used as a model of food allergy, whereas RagD10 mice are a cross between DO11.10 mice and *Rag2* knockout mice and are used as a model of oral tolerance. When fed a diet containing egg white (EW) protein, the most abundant component of which is OVA, RagD10 mice acquire oral tolerance to the EW diet that is characterized by induction of OVA-specific Tregs expressing Foxp3, a master regulator of Tregs, and a quick decline of immune responses. The mice predominantly produce IFN-γ by stimulation with anti-CD3 antibody or OVA, indicating Th1-biased responses [22]. In contrast, Rag23−3 mice develop food-allergic enteropathy that is characterized by a lack of OVA-specific Foxp3$^+$ Tregs during the allergic inflammation phase and strong IL-4 responses, when they were stimulated with anti-CD3 antibody or OVA [22]. Although continuous EW-feeding increases the OVA-specific Foxp3$^+$ Treg population and Rag23−3 mice recover from the allergic inflammation, our previous findings indicate that these Tregs are unable to prevent the reoccurrence of food allergy [22], suggesting that continuous allergen administration might not be a suitable approach for obtaining stable Tregs. Indeed, in the same study, we showed that the stability of OVA-specific Tregs induced during feeding with EW-diet was higher in DO11.10 mice than in OVA23−3 mice [22].

It has been reported that the functions and phenotypes of Tregs differentiated by using a combination of anti-CD3 and anti-CD28 monoclonal antibodies (mAbs) differ from those of Tregs differentiated with an antigen in the presence of antigen-presenting cells [24]. Thus, in the present study, we examined how to obtain stable and suppressive Tregs from the Rag23−3 food-allergic enteropathy model by comparing both the inducing means and their resultant differentiation levels and functions of Tregs induced in Rag23−3 and RagD10 mice when fed an EW diet. We found that OVA-specific Tregs differentiated from naïve T cells from the mesenteric lymph nodes (mLNs) and that, when activated under appropriate conditions using Abs, these OVA-specific Tregs prevented allergic inflammation, even when the cells were from the food-allergic model (EW-fed Rag23−3 mice). However, we also found that a severe allergic cytokine milieu, characterized by prominent production of IFN-γ and interleukin (IL)-4 by CD4$^+$ T cells, prevented Treg induction, even in CD4$^+$ T cells from the tolerant model (RagD10 mice).

## Materials and methods

### Animals

Rag23−3 mice (OVA23−3 mice [25] crossed with *Rag2* knockout mice) were kindly provided by S. Habu (Tokai University School of Medicine, Kanagawa, Japan). RagD10 mice (DO11.10 mice crossed with *Rag2* knockout mice) were kindly provided by Y. Wakatsuki (Kyoto University, Kyoto Japan). Both strains of mouse carried the BALB/cA background, possessed OVA-specific T-cell-receptor genes, and were bred at Sankyo Labo Service Corporation Inc. (Tokyo, Japan). The only lymphocytes that these strains of mice produce in response to OVA challenge are OVA-specific T cells; therefore, we can observe the antigen-specific T cell responses without consideration of B cell responses. When fed a diet containing EW protein for 9 days, RagD10 mice develop tolerance to the protein, whereas Rag23−3 mice show a severe allergic response and do not develop tolerance. In all experiments, sex-matched mice older than 8 weeks of age were used [22].

The mice were housed in cages (≤5 mice/cage; 182×260×128 mm; CL-0103–2; CLEA Japan, Inc.) at The University of Tokyo (Tokyo, Japan) and were maintained under specific-pathogen-free conditions at a room temperature of 22°C and under a 12:12-h light:dark cycle. Sterilized deionized drinking water and sterilized commercial chow were provided *ad libitum*. The maintenance conditions were monitored daily. When necessary for cellular and histological analysis, mice were euthanized by cervical dislocation by experts. During the adoptive transfer of Tregs, to facilitate the subsequent injection of cells into the caudal vein, a mouse was placed in a small box with only its tail sticking out through the hole in the box

for a few minutes. The injection was performed without anesthetization as smoothly and quickly as possible by experts. All experiments were performed in accordance with the guidelines of the University of Tokyo for the care and use of animals.

## Antigen administration

Antigen (OVA) was administered to the mice via their feed. The antigen-containing diet was a solid feed with the protein fraction consisting only of EW (Funabashi Farm Co., Chiba, Japan). A control diet was also used, which was a solid feed with the protein fraction consisting only of casein (CN, Funabashi Farm). The mice were fed one of these diets *ad libitum* for 7 days.

## Cell preparation and sorting

To prepare single-cell suspensions from mLNs (whole cells), mLNs were collected and cut into small pieces in complete RPMI 1640 medium (Thermo Fisher Scientific, Rockford, IL) supplemented with fetal calf serum (FCS; 10%, Thermo Fisher Scientific), penicillin/streptomycin (100 U/mL, Thermo Fisher Scientific), and 2-mercaptoethanol (50 µM, Fujifilm Wako Pure Chemical Corp, Osaka, Japan). The small pieces of mLN were cultured in complete RPMI1640 medium supplemented with 10 mM HEPES (Thermo Fisher Scientific), 1 mg/mL collagenase (final concentration; Fujifilm Wako Pure Chemical Corp), and 1 mg/mL DNase I (final concentration, Roche, Mannheim, Germany) with stirring at 37°C for 70 min. After incubation, the resulting mixture was passed through a 100-µm cell strainer (Falcon, Corning, NY, USA), and the cells collected through the strainer were washed twice and then resuspended in the complete RPMI. To prepare single-cell suspensions from spleen (whole cells), spleens were harvested, ground in complete RPMI medium with the bottom of a plastic syringe piston (Terumo, Tokyo, Japan), and passed through an 86-µm Tetron mesh (Tokyo Screen, Tokyo). The cells that passed through the mesh were washed twice and resuspended in complete RPMI medium. CD4$^+$ T cells, CD44$^{lo}$CD62L$^{hi}$CD4$^+$ T cells (naïve T cells), and effector/effector memory CD44$^{hi}$CD62L$^{lo}$CD4$^+$ T cells (effector/effector memory; EMTs) were isolated from the single-cell suspensions obtained from the mLNs and spleens of Rag23−3 mice fed with the CN- or EW-diet for 7 days. CD4$^+$ T cells were isolated by using a magnetic-activated cell sorting system (Miltenyi Biotec, Bergisch Gladbach, Germany) in accordance with the manufacturer's instructions. For cell sorting, the single cells (4–8 × 10$^7$ spleen or mLN cells) were suspended in 1 mL of phosphate-buffered saline containing 2% FCS (2% FCS–PBS) and then blocked by incubation with an anti-CD16/32 mAb (BD Pharmingen, Franklin Lakes, NJ, USA) at 4°C for 15 min. Next, the cell suspensions were centrifuged at 1400 rpm and 4°C for 5 min, resuspended in 1 mL of 2% FCS–PBS, and stained with allophycocyanin (APC)-conjugated anti-mouse CD4 (GK1.5; BioLegend, San Diego, CA, USA), APC-Cy7-conjugated anti-mouse CD44 (IM7; BD Pharmingen), and phycoerythrin (PE)-Cy7-conjugated anti-mouse CD62L (MEL-14; BioLegend) mAbs at 4°C for 20 min. After washes with 2% FCS–PBS and resuspension in 1 mL of 2% FCS–PBS, the cells were stained with 7-aminoactinomycin D (BioLegend) at room temperature for 5 min. After washes, the cells were resuspended in 2 mL of 2% FCS–PBS (2–4 × 10$^7$ cells/mL), and the naïve T cells and EMTs were sorted by using a FACSAria II cell sorter (BD Biosciences, Franklin lakes, NJ, USA).

## Treg-polarization culture

To induce Foxp3$^+$ Tregs, whole cells (2 × 10$^5$ cells/well in 96-well plates), CD4$^+$ T cells (1 × 10$^5$ cells/well in 96-well plates or 1 × 10$^6$ cells/well in 24-well plates), naïve cells (1 × 10$^5$ cells/well in 96-well plates), or EMTs (1 × 10$^5$ cells/well in 96-well plates) were incubated under *in vitro* Treg-polarization culture conditions in the presence of transforming growth factor-beta (TGF-β) 1 (2 ng/mL; R&D Systems, Minneapolis, MN, USA), retinoic acid (1 µM; Fujifilm Wako Pure Chemical, Osaka, Japan), and recombinant (r) IL-2 (2 ng/mL; R&D Systems) for 48 or 72 h. Whole cells were stimulated with OVA (0.25 mg/mL; Sigma-Aldrich, St. Louis, MO, USA), whereas CD4$^+$ T cells, naïve T cells, and EMTs were stimulated with plate-bound anti-CD3 (145-2C11; BD Biosciences) and anti-CD28 (37.51; BioLegend) mAbs. In some experiments,

pooled whole cells or purified CD4+ T cells were cultured with plate-bound anti-CD3 and anti-CD28 mAbs and rIL-2 (2 ng/mL) as a control condition. The culture supernatant obtained from each culture was also used for analysis of cytokine production by enzyme-linked immunosorbent assay (ELISA).

## Flow cytometry

Flow cytometry was used to identify the phenotype of CD4+ T cells by analyzing the expression of cell-surface molecules. Cells collected from the Treg-polarization culture were incubated with anti-CD16/32 mAb (BD Pharmingen) at 4°C for 15 min to block non-specific binding to Fc receptors. After washes, the cells were stained with the following fluorescent mAbs: fluorescein isothiocyanate (FITC)-conjugated anti-mouse CD4 mAb (H129.19; BD Pharmingen), APC-conjugated anti-mouse CD4 mAb (GK1.5; BioLegend), FITC-conjugated anti-mouse CD25 mAb (3C7; BioLegend), PE/Cy7-conjugated anti-mouse IL-4 receptor (IL-4R) mAb (1015F8; BioLegend), biotin-conjugated anti-mouse IFN-γ receptor (IFN-γR) mAb (2E2; BioLegend), anti-APC-Cy7-conjugated anti-mouse CD44 mAb (IM7; BD Pharmingen), and PE-Cy7-conjugated anti-mouse CD62L mAb (MEL-14; BioLegend). Foxp3 intracellular staining was performed by using an eBioscience Foxp3/Transcription Factor Staining Set (Invitrogen). Briefly, cells were fixed, permeabilized, and stained with APC- or PE-conjugated anti-mouse Foxp3 mAb (FJK-16s; eBioscience, Santa Clara, CA, USA). A FACSVerse cell analyzer (BD Biosciences) and FlowJo software (Ver 10.7.1, BD Biosciences) were used for analysis.

## ELISA

The amounts of IL-2, IL-4, and IFN-γ in culture supernatants were measured by ELISA, as described previously [23]. Anti-mouse IL-2 (JES6-1A12; BD Biosciences), anti-mouse IL-4 (11B11; BD Biosciences), and anti-mouse IFN-γ (R4-6A2; BD Biosciences) mAbs were used as capture Abs; biotin-conjugated anti-mouse IL-2 (JES6-5H4; BD Biosciences), biotin-conjugated anti-mouse IL-4 (BVD6-24G2; BD Biosciences), and biotin-conjugated anti-mouse IFN-γ (XMG1.2; BD Biosciences) mAbs were used as detection Abs. Recombinant murine IL-2 (rIL-2; Peprotech, Rocky Hill, NJ, USA), IL-4 (rIL-4; R&D Systems), and IFN-γ (rIFN-γ; Peprotech) were used as standards. The amount of IL-10 production in culture supernatant was analyzed by using Mouse IL-10 Quantikine ELISA kit (R&D Systems).

## Induction of Tregs by using CD4+ T cell culture supernatants

To prepare the supernatants, RagD10 and Rag23−3 mice were fed the EW or CN diet for 7 days and spleens were harvested and processed into single-cell suspensions. Then, CD4+ T cells (1 × 10^5 cells in 96 well plates) isolated from the suspensions were stimulated with plate-bound anti-CD3 (BD Biosciences) and anti-CD28 (BioLegend) mAbs for 48 h, and the supernatant was collected. To analyze the effects of cytokines contained in the supernatants on the induction of Tregs, CD4+ T cells from spleens of untreated RagD10 mice were stimulated with plate-bound anti-CD3 (BD Biosciences) and anti-CD28 (BioLegend) Abs and then cultured with the prepared supernatants under Treg-polarization culture conditions for 72 h. Each culture supernatant of the cells derived from CN- or EW-fed RagD10 or Rag23−3 mice described above was added to 50% of the total volume of the Treg-polarization culture medium of the cells from the untreated RagD10 mice. To identify cytokines inhibiting Treg induction from CD4+ T cells of untreated RagD10 mice, the supernatant derived from EW-fed R23-3 mice was added to the culture medium with or without anti-IL-4 Ab (αIL-4 Ab; 1 μg/mL; BioLegend) or anti-IFN-γ Ab (αIFN-γ Ab; 1 μg/mL; BioLegend) and cultured for 72 h. The volume of the supernatant, or the medium with or without Abs, was 25% of the total culture medium (200 μL) in each well.

## Treg induction under rIL-4 or rIFN-γ supplemented conditions

CD4+ T cells (1 × 10^5 cells in 96-well plates) were prepared from splenocytes derived from Rag10 mice fed the CN-diet for 7 days and then stimulated with plate-bound anti-CD3 and anti-CD28 Abs under Treg-polarization conditions

supplemented with mouse rIFN-γ (0, 1.0, 2.0, or 3.5 ng/mL; 315−05, Peprotech) and mouse rIL-4 (0, 0.5, 1.5, or 2.5 ng/mL; 214−14, Peprotech) for 72 h.

## Adoptive transfer of Tregs into Rag23-3 mice

CD4+ T cells isolated from whole-cell suspensions prepared from the spleens of untreated Rag23−3 mice ($1 \times 10^6$ cells) were cultured under control conditions by being stimulated with plate-bound anti-CD3 and anti-CD28 mAbs in the presence of 2 ng/mL rIL-2, or cultured under Treg-polarization conditions by being stimulated with plate-bound anti-CD3 and anti-CD28 mAbs in the presence of rIL-2 (2 ng/mL), TGF-β1 (5 ng/mL), and retinoic acid (1 µM) in 24-well plates. Using the FlowJo software, we confirmed that the purity of Foxp3+ cells was > 90% in the Treg-polarization medium. Control cells and Treg-abundant cells were collected from each culture, re-suspended in PBS, and transferred via a caudal vein to untreated Rag23−3 mice ($1 \times 10^7$ cells/head). As a positive control group, untreated Rag23−3 mice were injected with PBS. From 1 day after the injection of the cells or PBS, each group of Rag23−3 mice were fed the EW diet for 8 days and their weights were measured. On day 7 of EW feeding, the mice were euthanized. Jejunum was collected for histological analysis, and mLN and spleen were collected for analysis of CD4+ T cell proliferation and their cytokine (IL-4 and IFN-γ) production.

## Histological analysis

A 3-cm section of jejunum, 10 cm away from the distal end of the stomach, was harvested and fixed in 10% formalin (Fujifilm Wako Pure Chemical). Tissues were embedded in paraffin, 4-µm section were obtained, and the sections were stained with hematoxylin and eosin. Morphological features were observed under a BX-51 microscope (Olympus, Tokyo, Japan).

## Measurement of cytokine production by CD4+ T cells isolated from Rag23−3 mice that received adoptive transfer of Tregs

CD4+ T cells were isolated by magnetic-activated cell sorting from single-cell suspensions of mLN or spleen from the recipient mice used in the adoptive Treg transfer experiment. The CD4+ T cells ($1 \times 10^5$ cells/well) were then stimulated with OVA (0.25 mg/mL; Sigma-Aldrich) plus splenocytes from BALB/cA mice (CLEA Japan Inc., Tokyo, Japan) treated with mitomycin C (Fujifilm Wako Pure Chemical Corp) as antigen-presenting cells (CD4+ T cell-culture) for 48 h. The amount of IL-4 or IFN-γ in the culture supernatant was measured by ELISA, as already described.

## Proliferation assay

CD4+ T cells were stimulated with OVA and antigen-presenting cells for 24 h and then [3H]-thymidine (0.5 µCi/well; Moravek Biochemicals, Brea, California, USA) was added to each well. After incubation for 16 h, cells were harvested and the incorporation of [3H]-thymidine was measured by a scintillation counter (Perkin Elmer, Rodgau, Germany).

## Statistical analysis

Unpaired student's t-test was performed for comparison between two groups. To compare more than two groups of data, Tukey's honestly significant difference (HSD) test or Dunnett's test was performed following ANOVA by using R software (Ver 3.3.1). $p$ values below 0.05 were considered significant. All data in the graphs are shown as mean or mean ± 1 SD.

## Ethics statement

This study was carried out in strict accordance with the recommendations in the Guide for the Animal Use Committee of the Faculty of Agriculture at the University of Tokyo. Approval numbers: P15-022, P15-023, P17-044, and P19-038. All animal experiments were conducted at the University of Tokyo according to the ARRIVE guidelines regarding the care

                                                                                    

and use of experimental animals (https://www.u-tokyo.ac.jp/adm/lifescience/ja/doukisoku.html). Our experiments were approved without setting preemptive humane endpoints, because during the experimental period (7 days), the mice did not exhibit other clinical signs than weight loss.

## Results

### Weight change and intestinal histology in Rag23−3 and RagD10 mice

In our previous study using the Rag23−3 and RagD10 mice fed for 7 days with either the EW or CN diet, we reported that EW-fed Rag23−3 mice have a difficulty in inducing Foxp3 expression by CD4⁺ T cells, whereas EW-fed RagD10 mice readily induce the expression, which resulted in intestinal inflammation and weight loss in the Rag23−3 but not the RagD10 mice [22]. In the present study, we observed results consistent with those of our previous study, with the EW-fed Rag23−3 mice showing a greater loss of body weight compared with the CN-fed Rag23−3 mice or EW- or CN-fed RagD10 mice (S1A Fig). Hematoxylin and eosin staining of sections of jejunum revealed that the EW-fed Rag23−3 mice also developed intestinal inflammation during the experimental period, whereas the other mice did not (S1B Fig).

### Stimulation with OVA impairs Treg induction but promotes EMT induction and IL-4 and IFN-γ production in Rag23−3 mice, but not in RagD10 mice

To investigate the abilities of the CD4⁺ T cells in the two mouse models to differentiate into Tregs, we firstly stimulated whole cells isolated from mLN cells (including OVA-specific T cells and antigen-presenting cells) with OVA under Treg-polarization culture conditions and examined by flow cytometry; the gating strategy used to identify the Treg and EMT subpopulations within the CD4⁺ T cell population is shown in Fig 1A. The Treg population was significantly larger in the EW-fed RagD10 mice than in those fed the CN-diet, accounting for around 22% and 5% of the CD4⁺ T cell population, respectively (Fig 1B). In contrast, the sizes of the Treg populations were comparable in the CN- and EW-fed Rag23−3 mice, with both also around 5%. The EMT populations in both mouse strains fed the EW-diet were significantly larger than those in the mice fed the CN-diet (RagD10: around 8% vs. 3%; Rag23−3: around 20% vs. 5%; Fig 1C). In both treatment groups for both mouse strains, around 80% of the Tregs expressed CD25, a common marker of murine Tregs and a component of the Treg IL-2 receptor (Fig 1B). The frequency of CD25⁺ T cells significantly increased in EW-fed mice in spite of the difference of the strains, showing that EW feeding even at low levels, induced Tregs, even at minor level under the severe allergic condition of Rag23−3 mice.

Examination of the cytokine milieu produced by the mLN cells after stimulation with OVA revealed characteristic cytokine profiles for IL-4 production, a marker of Th2-type immune responses; IFN-γ production, a marker of Th1-type immune responses; and IL-2 production, a marker of T cell proliferation (Fig 1D). The RagD10 mice, irrespective of diet, showed no detectable IL-4 production, high IFN-γ production, and no IL-2 production. In contrast, the Rag23−3 mice fed the EW-diet showed high IL-4 production, whereas those fed the CN-diet showed no IL-4 production. For IFN-γ production, the EW-fed Rag23−3 mice showed the same high production as was observed for the RagD10 mice, but the CN-fed mice showed significantly less production. For IL-2, the Rag23−3 mice showed some production irrespective of diet, but the production in the CN-fed mice was significantly greater than that in the EW-fed mice.

Taken together, these results indicate that OVA stimulation resulted in greater T cell proliferation in the Rag23−3 mice than in the RagD10 mice, which is consistent with our previous findings [22]. In addition, these results suggest that the CD4⁺ T cell population in EW-fed Rag23−3 mice is considerably activated and that EMTs are promoted from the naïve to the Th2 phenotype. Given the observed increase of Treg frequency in the EW-fed RagD10 mice, we expected to also observe increased IL-2 production in these mice, but this was not found to be the case. Since these data were collected at 7 days after the start of EW feeding, these findings imply that the response to OVA in RagD10 mice is fully attenuated by day 7 of EW feeding. Although IFN-γ production tended to be suppressed by EW-feeding compared with those in CN-fed

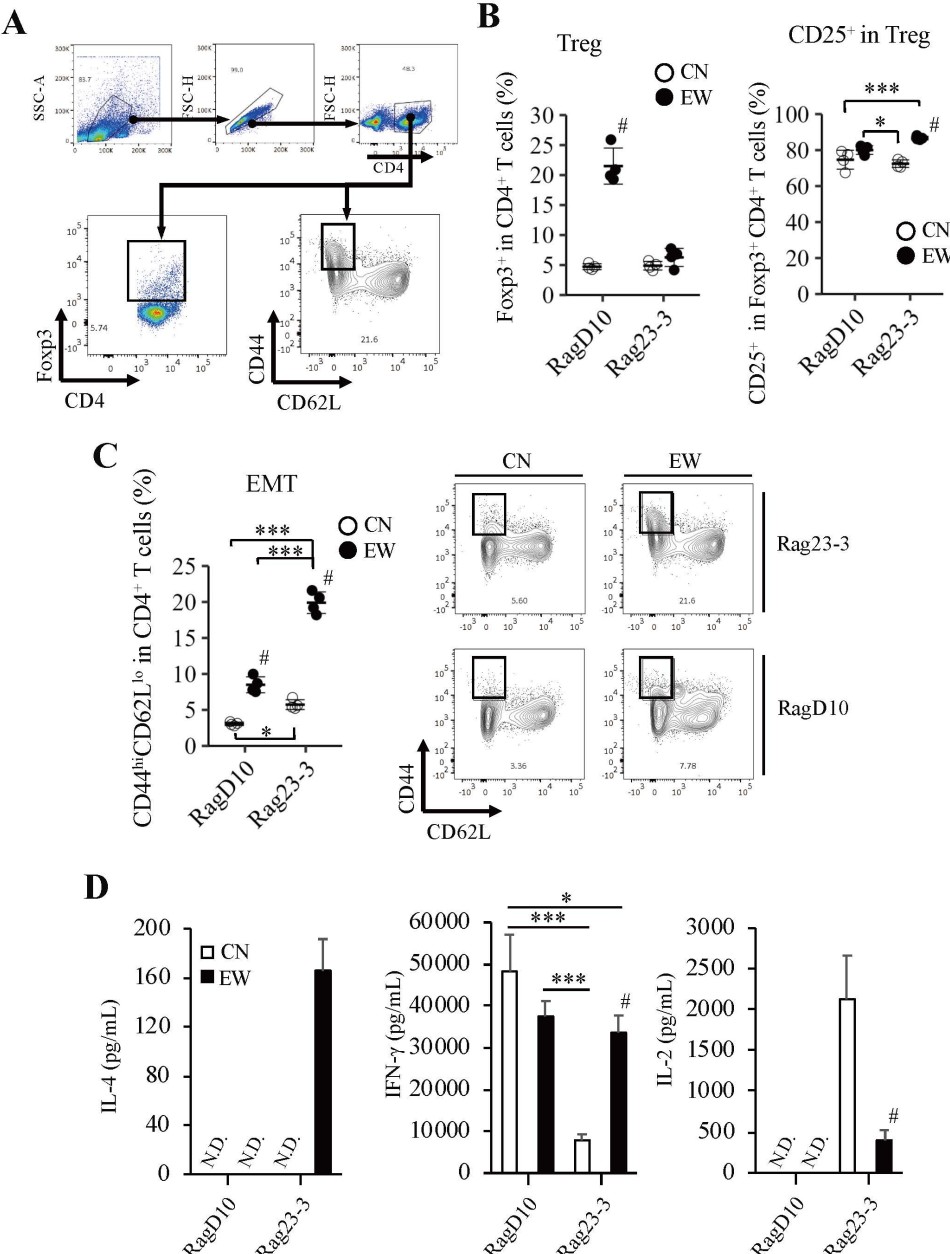

**Fig 1. Ovalbumin (OVA) stimulation of mesenteric lymph node T cells impaired regulatory T cell differentiation under a characteristic cytokine milieu accompanied by greater differentiation of effector/effector memory T cells (EMTs) in Rag23−3 mice, but not RagD10 mice.** RagD10 mice and Rag23−3 mice were fed a diet containing egg white (EW) or casein (CN) for 7 days. Whole cells isolated from the mesenteric lymph nodes (mLNs) were cultured under Treg-polarization culture conditions and stimulated with OVA. A) Gating strategy used to identify the regulatory T cell (Treg; Foxp3+) and effector/effector memory T cell (EMT; CD44hiCD62lo) subpopulations within the CD4+ T cell population. B) Frequency of Tregs in the CD4+ T cell population and of CD25 expression in the Treg population. C) (left) Frequency of EMTs in the CD4+ T cell population. Each plot in b) and c-left) represents the value for a single well and the horizontal lines indicate the mean value (n = 4, mixture of cells from two to three mice/group). (right) Flow cytometric analysis of the EMTs of Rag23−3 and RagD10 mice. EMT fractions are shown within the black squares. D) Concentrations of interleukin (IL)-4, interferon gamma (IFN-γ), and IL-2 in the culture supernatants (n = 3, mixture of cells from three mice/group). Data are representative of two independent experiments. Statistical analysis: Tukey's HSD test [#$p < 0.05$ (CN vs EW in each strain); * $p < 0.05$, ** $p < 0.01$, and *** $p < 0.001$ (between different groups)]. N.D., not detected.

RagD10 mice, the RagD10 mice showed significantly greater IFN-γ production, indicating that the immune response to OVA in RagD10 mice is fundamentally of the Th1 type, which is consistent with previous reports [22,26]

The T cells of CN-fed Rag23−3 mice are reported to produce large amounts of IL-2, a cytokine needed for Treg differentiation, but not IL-4, which is suggested to suppress Treg induction [27]. These findings are inconsistent with our finding that the Treg population is very small in the CN-fed Rag23−3 mice (Fig 1B) despite high IL-2 production (Fig 1D). There are two possibilities for this inconsistency: 1) excessive activation of EMTs inhibit Treg differentiation, independently of IL-2 production [28] or 2) excess activation of the overall immune response through antigen-specific T cell receptor via antigen presenting cells and resultant suppression of their TGF-β1 receptor expression on T cells reduce the responsiveness of CD4+ T cells to TGF-β1, leading resultant inhibition of Treg differentiation [29]. However, the Treg population induced by responses of CD4+ T cells to TGF-β1 was rather larger in Rag 23−3 mice than in RagD10 mice (S2 Fig).

## Antibodies stimulation induces CD4+ T cells from CN-fed, but not EW-fed Rag23−3 mice to differentiate into Tregs, whereas CD4+ T cells from RagD10 mice differentiate regardless of diet

The results presented in the previous section suggested that transcription factor activation via antigen-specific T cell receptors signaling for Foxp3 expression may differ in the presence (antigen plus antigen-presenting cell) or absence (no-antigen mediated) of OVA as reported by Zhao C. et al [24]. Therefore, we next examined the effects of no-antigen mediated induction of Tregs and EMTs in CD4+ T cells isolated from the spleens and mLNs of the model mice by using anti-CD3 and anti-CD28 mAbs. In contrast to the earlier data, the frequency of Tregs in the T cells isolated from the spleen or mLNs of CN-fed Rag23−3 mice was surprisingly comparable to those in T cells of CN-fed RagD10 mice (Fig 2A). However, the frequency of Tregs in the EW-fed Rag23−3 mice was significantly reduced compared with that in the CN-fed mice for both the spleen-derived and mLN-derived cells, furthermore is higher in the spleen derived compared with mLN-derived cells, which was consistent with the data presented in Fig 1. In RagD10 mice, the frequency of Tregs in the mLN-derived cells was significantly higher in the EW-fed mice than in the CN-fed mice, whereas that in the cells from the spleen was significantly lower in the EW-fed mice than in the CN-fed mice. We have reported that Foxp3 molecule is more easily induced in the mLN than in the spleen, but in the food-allergic intestinal inflammation model like Rag23−3 or OVA23−3 mice, this is not necessarily the cases [22]. This is because CD4+ T cells from the mLN of Rag23−3 mice produce high levels of IL-4 in response to OVA that invade into the intestinal tract after consumption of the EW diet, preventing the differentiation of CD4+ T cells into Tregs [30]. Tregs induced in RagD10 mice responded normally against antigen entry in the intestine as indicated by Traxinger, et al. [31]. In contrast, in the spleen, the number of CD4+ T cells is less and the level of IL-4 production is lower than those in mLN [30], which lead to higher induction of Tregs compared with the induction in mLN. In the present study, almost 100% of the induced Tregs in both models and with both diets expressed CD25 (Fig 2A), which is similar to what was observed for Tregs induced from mLN cells by antigen stimulation (Fig 1B).

In both the spleen- and mLN-derived cells, the frequency of EMTs was significantly higher in the EW-fed Rag23−3 mice than in those fed the CN-diet (Fig 2A). The frequency in EW-fed Rag23−3 mice was significantly higher than that in the CN- and EW-fed RagD10 mice in both tissues. In addition, the frequency in CN-fed Rag23−3 mice was significantly lower than those in the CN- and EW-fed RagD10 mice for both tissues, indicating that Ab-mediated signaling inhibited differentiation of naïve CD4+ T cells into EMTs, even in the absence of an antigen.

Large amounts of IL-4 and IFN-γ were detected in the supernatants of the CD4+ T cells isolated from the spleen and mLN of EW-fed Rag23−3 mice, but not in the other three experimental groups, which showed no production (Fig 2B). This indicates the presence of excessive activation of CD4+ T cells in the EW group compared with the other groups IL-2 production was observed in all groups and was comparable for all groups for the cells derived from the spleen. However, IL-2 production was significantly (EW) or tended to be (CN) higher in the cells isolated from the mLN of the Rag23−3 mice than in those from the RagD10 mice. The level of IL-10 production in the spleen cells and in the mLN cells was significantly higher in the Rag23−3 mice than in the RagD10 mice when fed the EW-diet. The amount of IL-10 produced by CD4+

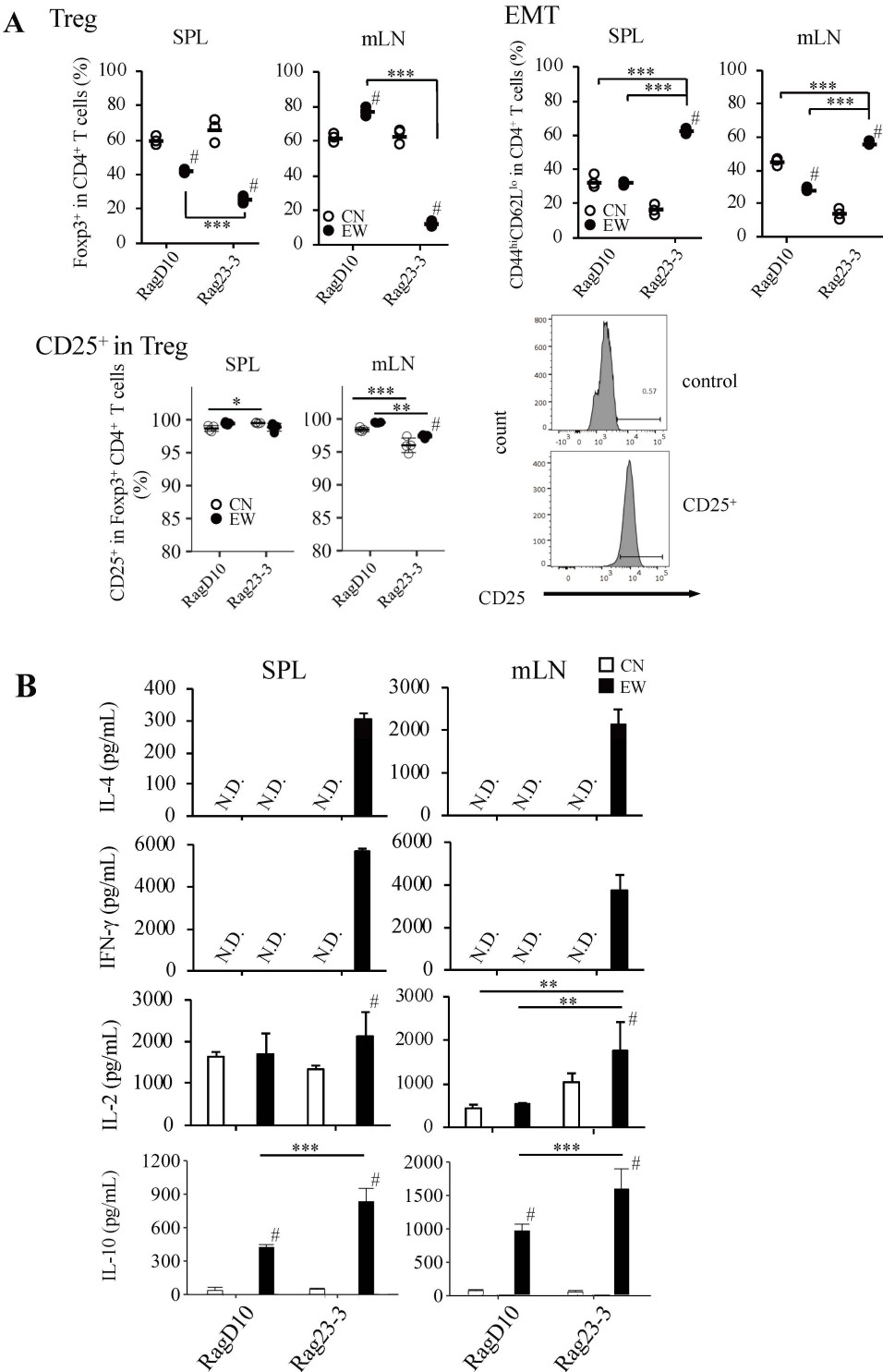

**Fig 2. Antibody (Ab)-mediated stimulation differentiated CD4 [+] T cells into regulatory T cells in casein (CN) diet-fed Rag23−3 like that that did in RagD10 mice.** RagD10 and Rag23−3 mice were fed diets containing egg white (EW) or casein (CN) for 7 days. CD4[+] T cells (1 × 10[5]) from spleen (SPL) and mesenteric lymph nodes (mLN) were stimulated with plate-bound anti-CD3 and anti-CD28 mAb and cultured under Treg-polarization culture conditions. A) (Top) Frequencies of regulatory T cells (Tregs; Foxp3[+]) and effector/effector memory T cells (EMT; CD44hiCD62Llo) in the CD4[+] T cell populations isolated from the spleen (SPL) and mesenteric lymph nodes (mLN). (Bottom) Frequencies of CD25[+] cells in the Treg population and the

associated flow cytometry histogram. Each plot indicates the value for an individual well and the horizontal bars indicate the mean values (n = 3, mixture of cells from three mice/group). B) Levels of interleukin (IL)-4, interferon gamma (IFN-γ), IL-2 and IL-10 in the culture supernatants (n = 3, mixture of cells from three mice/group). Data are representative of two independent experiments. Statistical analysis: Tukey's HSD test [#$p < 0.05$ (CN vs EW in each strain); *$p < 0.05$, **$p < 0.01$, and ***$p < 0.001$ (between different groups)]. N.D. = not detected.

T cells in mLN was much more than that in spleen EW-fed mice. These results suggest that oral administration of EW induced overactivation of intestinal immune responses, as shown by the production of IL-4, IFN-γ, and IL-10 in EW-fed Rag23−3 mice. In addition, Tregs differentiated from CD4+ T cells by stimulation with mAbs in both strains of EW-fed mice probably have sufficient regulatory function producing IL-10, but suppressive function of the Tregs in both strains of CN-fed mice may be independent of IL-10.

The finding that stimulation with OVA did not fully induce Treg differentiation in CN-fed Rag23−3 mice, but stimulation with mAbs did, suggests the presence of one or more factors that prevent Treg induction, such as a molecule expressed by signal transduction in CD4+ T cells via interaction of T-cell receptor with the antigen in the presence of antigen-presenting cells.

## Excessive IL-4 and IFN-γ production by CD4+ T cells from EW-fed Rag23−3 mice suppresses Treg differentiation of CD4+ T cells from RagD10 mice

Unlike in EW-fed RagD10 mice, the higher frequency of EMT induction in the EW-fed Rag23−3 mice was observed to have occurred concomitantly with greater cytokine production, suggesting that this higher cytokine production prevented the differentiation of CD4+ T cells into Tregs. To examine this further, we first examined whether the supernatants of CD4+ T cells isolated from the spleen of EW- or CN-fed Rag23−3 or RagD10 mice did indeed have the ability to prevent the differentiation of CD4+ T cells into Tregs. The cytokine profiles of the supernatants used in this experiment are shown in S3 Fig, and these profiles were almost consistent with those shown in Fig 2B. When CD4+ T cells from spleens of untreated RagD10 mice were stimulated with plate-bound anti-CD3 and anti-CD28 Abs and then cultured with the prepared supernatants under Treg-polarization culture conditions, the supernatant from the EW-fed Rag23−3 mice significantly suppressed the induction of Tregs compared with the no-treatment control (Fig 3A). Furthermore, when anti-IL-4 or anti- IFN-γ Abs were added to the supernatant from the EW-fed Rag23−3 mice, the differentiation of Tregs was partially restored (Fig 3B), confirming that IL-4 and IFN-γ production by EMTs inhibits the differentiation of CD4+ T cells into Tregs. The frequency of Tregs in the CD4+ T cell population was higher regardless of the mouse strain when the supernatants from the CN-fed mice were used (Fig 3A), with the supernatant from the CN-fed Rag23−3 mice inducing significantly more Tregs compared with that from CN-fed RagD10, suggesting that the greater amount of IL-2 in the former supernatant contributed to the greater Treg differentiation (S3 Fig), as mentioned in [16]. Neutralization of IL-4 in the supernatant significantly reduced IFN-γ production, whereas neutralization of IFN-γ in the supernatant did not affect the production of IL-4 (Fig 3C). However, while the decrease of IFN-γ concentration in the culture supernatant had no effect on the differentiation of EMTs and their activities compared with no Ab treatment, decreasing the concentration of IL-4 significantly reduced the differentiation of EMTs (Fig 3B), but the levels suppressing the Treg differentiation (Fig 3B) and inhibiting their expression of cytokines receptors (S4B Fig) are similar despite of either neutralized mAbs against the cytokines used. Almost Tregs induced were CD25+ (S4A Fig). These results suggest that in contrast to the equal contribution of these cytokines to impaired Treg induction, only IL-4, not IFN-γ is associated with EMT expansion. Therefore, because IFN-γ does not regulate EMT activities, excessive IL-4 production regulates IL-4 and IFN-γ production by EMT cells and their activities, leading to suppressed Treg differentiation with IL-4 alone or with IFN-γ produced by activated EMT through excessive IL-4 production. The excessive IFN-γ may promote further IL-4 production creating a feedback loop and inhibition of Tregs. Together, these findings indicate that CD4+ T cells of Rag23−3 mice, which are strongly pathogenic, can differentiate into Tregs in vitro if they are incubated with anti-CD3 and anti-CD28 mAbs but not if they are incubated with an allergen (in this case OVA)

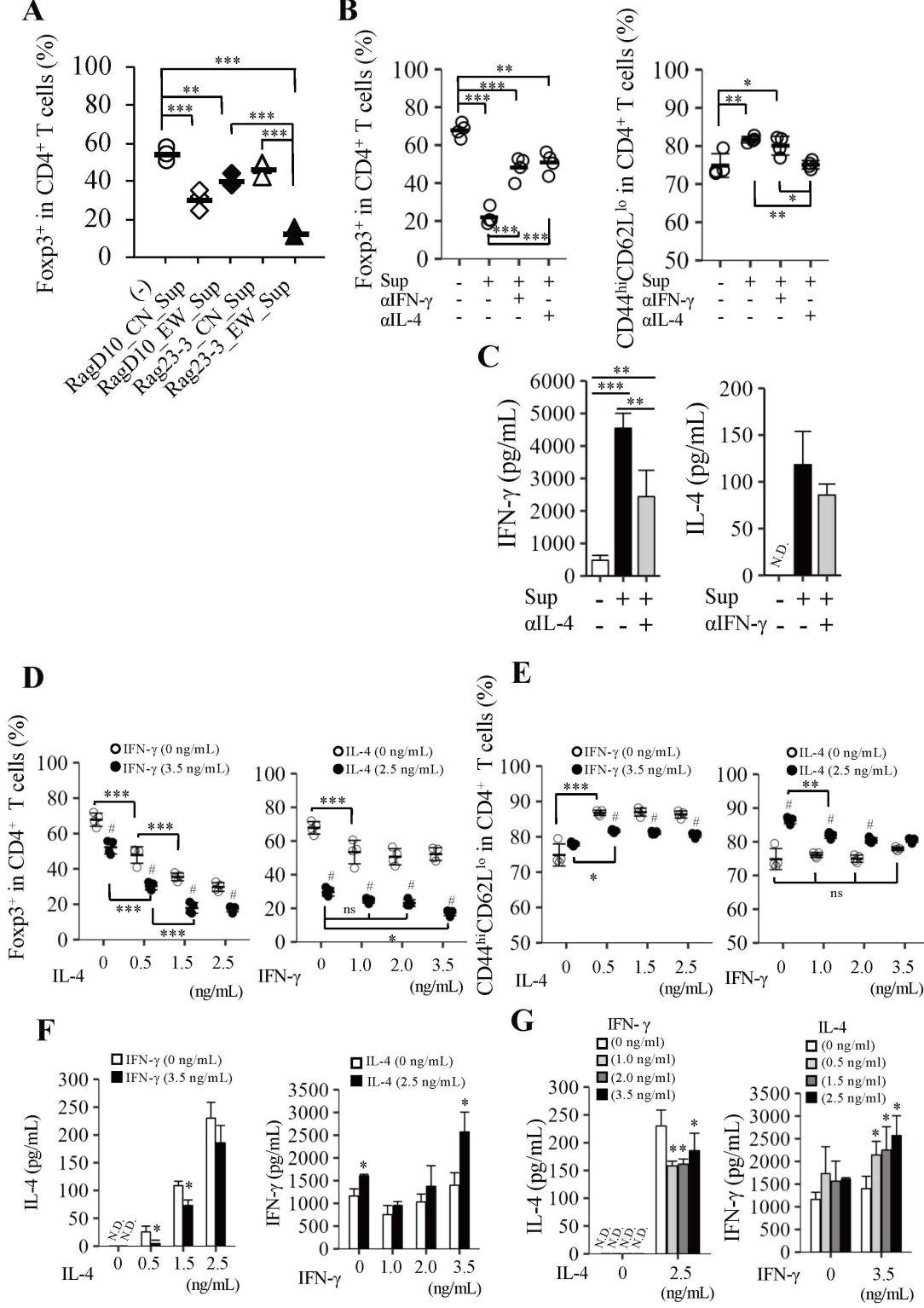

**Fig 3. Excessive production of interleukin (IL)-4 and interferon-gamma (IFN-γ) by CD4+ T cells from EW-fed Rag23−3 mice suppresses Treg differentiation of CD4+ T cells from RagD10 mice.** CD4+T cells were prepared from splenocytes of RagD10 and Rag23−3 mice fed a diet containing egg white (EW) or casein (CN) for 7 days. The cells were then stimulated with plate-bound anti-CD3 and anti-CD28 antibodies (mAbs) under

Treg-polarization culture conditions for 48 h, and the supernatants (Sup) were collected. A) Treg induction in spleen-derived CD4$^+$ T cells of untreated RagD10 mice cultured under Treg-polarization conditions for 72 h with plate-bound anti-CD3 and anti-CD28 mAbs with or without (-) the indicated supernatants (Sup). B) Regulatory T cell (Treg; Foxp3$^+$) and effector/effector memory T cell (EMT; CD44$^{hi}$CD62L$^{lo}$) induction in spleen-derived CD4$^+$ T cells from untreated RagD10 mice. They were cultured without the supernatant of EW-fed Rag23−3 mice (-) or cultured with the Sup with or without anti IL-4 or IFN-γ Abs, under Treg-polarization culture conditions for 72 h. C) IL-4 and IFN-γ concentrations in the culture supernatants obtained in (B). D, E) Frequencies of Treg (Foxp3$^+$) (D) or and EMT (CD44$^{hi}$CD62L$^{lo}$) (E) in CD4$^+$ cells prepared from splenocytes derived from RagD10 mice fed a diet containing casein for 7 days and stimulated with plate-bound anti-CD3 and anti-CD28 mAbs supplemented with rIFN-γ and/or rIL-4 under Treg-polarization culture conditions for 72 h. Each plot in A), B), D), and E) represents the value for a single well. Bar (-) indicates the mean values (n = 4, mixture of cells from two to three mice/group). F, G) IL-4 or IFN-γ concentrations in the culture supernatants obtained in (D and ES), respectively. (n = 4, mixture of cells from two to three mice/group). Data are representative of two independent experiments. Means ± SD. Analysis: Unpaired Student's t test or Dunnett's test were performed in Fig 3F and 3G respectively. *$p < 0.05$ [[IFN-γ (0 ng/mL) vs [IFN-γ (3.5 ng/mL)] or [IL-4 (0 ng/mL)] vs [IL-4 (2.5 ng/mL)]] in Fig 3F; *$p < 0.05$ [[IFN-γ (0 ng/mL) compared with [IFN-γ (1.0 ng/mL)], [IFN-γ (2.0 ng/mL)] and [IFN-γ (3.5 ng/mL)] or [IL-4 (0 ng/mL)] compared with [IL-4 (0.5 ng/mL)], [IL-4 (1.5 ng/mL)], and [IL-4 (2.5 ng/mL)]] in Fig 3G. Otherwise, Tukey's HSD test was performed [#$p < 0.05$ [IFN-γ (0 ng/mL) vs [IFN-γ (3.5 ng/mL)] or [IL-4 (0 ng/mL)] vs [IL-4 (2.5 ng/mL)] and *$p < 0.05$, ** $p < 0.01$, and *** $p < 0.001$ (between different groups)]. N.D., not detected.

or in an environment with excessive IL-4 and IFN-γ, like which were produced in the culture supernatant of mLN Tregs shown in Fig 2B (IL-4; 2.5 ng/mL, IFN-γ; 3.5 ng/mL).

Next, we further examined the contributions of the excessive IL-4 and IFN-γ to the promotion of EMT differentiation and inhibition of Treg differentiation. To do this, we added rIL-4 and/or rIFN-γ to Treg-polarization culture of CD4$^+$ T cells derived from the splenocyte population of CN-fed RagD10 mice. The Tregs induced were almost CD25$^+$ independently of the concentration of rIL-4 and/or IFN-γ (S4C Fig). Consistent with the results shown in Fig 3B, rIL-4 and rIFN-γ both inhibited Treg induction in dose dependent manner (Fig 3D). Under excessive IL-4 conditions [Fig 3D-right: 0 and 3.5 ng/mL of rIFN-γ in rIL-4 (2.5 ng/mL)], Treg differentiation was significantly less in the presence of excessive rIFN-γ compared than that in the absence of the protein, clearly indicating that IFN-γ reduces Treg differentiation even under not only without but also with excessive IL-4 condition. For EMT induction, while rIL-4 alone significantly promoted EMT induction, treatment with rIFN-γ alone neither promoted nor suppressed EMT induction (Fig 3E). However, when supplemented with rIL-4 (Fig 3E-left) to excessive IFN-γ condition, the IFN-γ significantly weakened EMT inducive effect of rIL-4 (Fig 3E-left). In addition, under excessive IL-4 conditions (Fig 3E-right), the frequency of EMTs also significantly decreased with rIFN-γ concentration in a dose dependent manner. These results suggest that excessive production of IFN-γ by EMT suppresses IL-4-mediated EMT activation.

Thus, we next assessed the production and receptor expression of these cytokines. The downregulated IL-4 production under condition of excess rIFN-γ (Fig 3F-left, Fig 3G-left) with no change in IL-4R expression (S4D Fig-lower left) suggests that EMT induction, which is caused by excessive IL-4, is impaired not via regulation of sensitivity to IL-4 but by downregulation of IL-4 production. In addition, excessive IL-4 enhance IFN-γR expression compared to no IL-4 (S4D Fig-upper-right), but the difference was not significant. rIL-4 promoted IFN-γ production regardless of rIFN-γ supplementation (Fig 3F-right, Fig 3G-right) in agreement with the hypothesis that excessive IL-4 initiates EMT induction, which then causes IFN-γ overproduction resultant inhibition of Treg differentiation, although IL-4 itself can suppress Treg differentiation (Fig 3D). In addition, excessive rIL-4 itself did not promote (S4D-lower-right) or rather downregulated (S4D Fig-lower-left) IL-4R expression. However, by supplementation with excessive rIFN-γ, IL-4R expression was enhanced, suggesting that in the activation producing excessive IL-4 and IFN-γ production, EMTs raised to increase CD4$^+$ T cell sensitivity to both cytokines most likely not via IFN-γR but rather IL-4R expression, resulting in increased Treg suppression.

### Adoptive transfer of Ab-stimulated Tregs from untreated Rag23−3 mice to EW-fed Rag23−3 mice have sufficient regulatory activity to prevent allergic inflammation

We proved in Fig 2 that naïve CD4$^+$ T cells had similar ability to differentiate into Tregs in Rag23−3 mice compared with those in RagD10 mice. Therefore, we next investigated whether the Tregs induced from Rag23−3 CD4$^+$ T cells possess

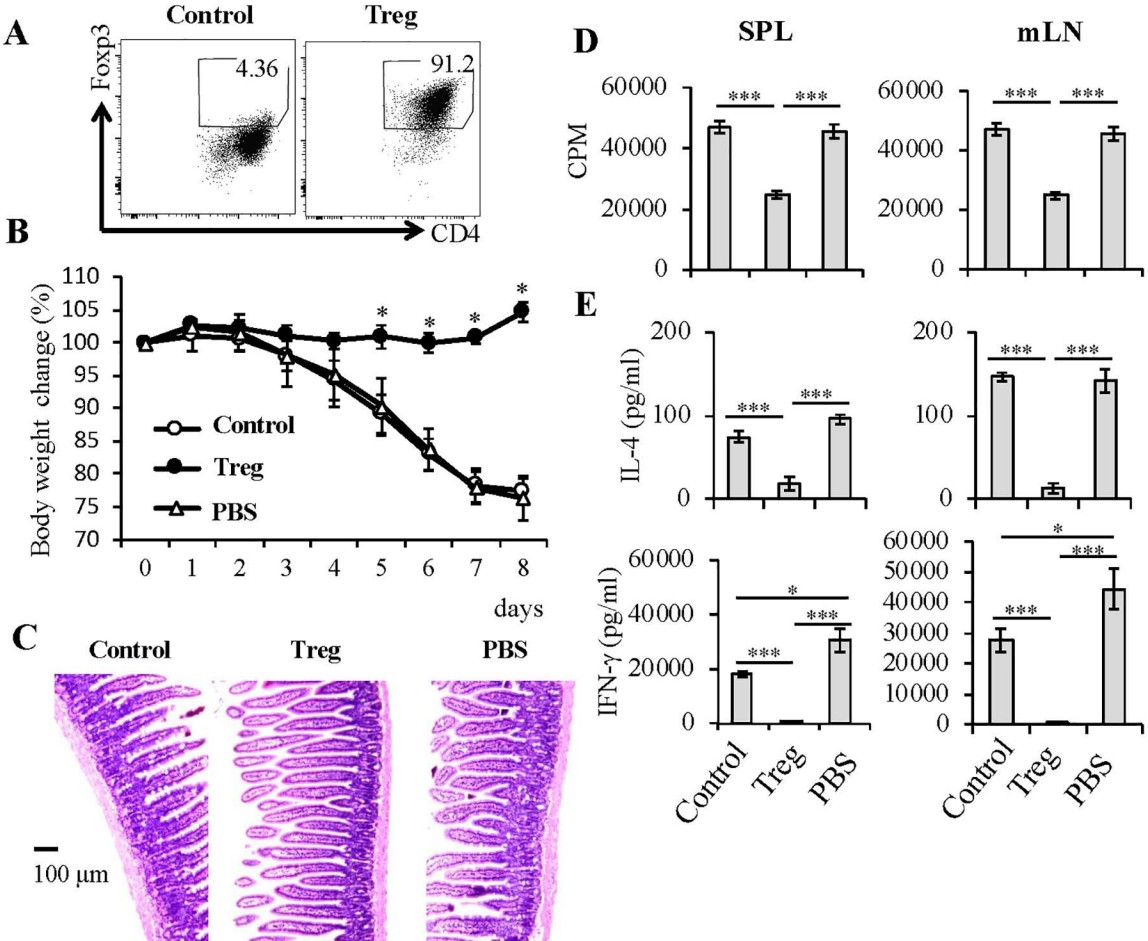

**Fig 4. Adoptive transfer of antibody (Ab)-stimulated Tregs from untreated Rag23−3 mice to EW-fed Rag23−3 mice have sufficient regulatory activity to prevent allergic inflammation.** A) Regulatory T cell (Treg; Foxp3+) population within the transferred CD4+ T cell population. Transferred CD4+ T cells were derived from spleen of untreated Rag23−3 mice cultured for 72 h under control conditions (Control; stimulated with plate-bound anti-CD3 and anti-CD28 mAbs and then incubated with recombinant IL-2 [rIL-2]) or under Treg-polarization conditions (Treg). After collection, the cells were intravenously injected into untreated Rag23−3 mice ($1 \times 10^7$ cells/mouse). As a control without cell transfer, Rag23−3 mice received phosphate-buffered saline (PBS). B) Time course of body-weight changes relative to initial values (100%). Weight changes were compared between the Treg and control groups each day. C) Hematoxylin and eosin–stained jejunum. Scale bars, 100 µm. D) [H³]-thymidine uptake (in counts per minute; CPM) of spleen (SPL)- or mesenteric lymph node (mLN)-derived CD4+ T cells isolated from the indicated treatment groups after egg white feeding and stimulated with OVA and antigen-presenting cells. E) Concentrations of interleukin (IL)-4 and interferon gamma in the supernatant of SPL or mLN-derived CD4+ T cells isolated from the indicated treatment groups after egg white feeding and incubation for 48 h. Data are representative of two or three independent experiments. Statistical analysis: Tukey's HSD test [*$p < 0.05$ (CN vs. each of the two groups) in Fig 4B; *$p < 0.05$, **$p < 0.01$, and ***$p < 0.001$ (between different groups) in Fig 4D and 4E].

sufficient regulatory activity to inhibit intestinal allergic responses. Spleen-derived CD4+ T cells from untreated Rag23−3 mice were cultured under control conditions or under Treg-polarization conditions. Cells from each culture were collected and intravenously injected into untreated Rag23−3 mice. Flow cytometry analysis of the cultured cells before the adoptive cell transfer showed that 91.2% of the Tregs were Foxp3+, whereas only 4.36% of the control cells were (Fig 4A). From one day after administration of either the control cells, Tregs, or PBS to untreated Rag23−3 mice, the mice were fed the EW diet for 8 days. The mice that received the control treatments (administered control cells or PBS; Control or PBS) showed a significantly lower body weight from day 5 compared with the mice that received the Tregs (Fig 4B). Histological

analysis of the jejunum showed a normal appearance in the Treg group, but morphological changes (i.e., crypt elongation, goblet cell hyperplasia, and cell infiltration into the villus) throughout the tissue in the control groups (Fig 4C). The body weight and histological data indicate that the transfer of the Tregs inhibited the induction of food-allergic enteropathy. On day 8 of EW feeding, spleen- and mLN-derived CD4+ T cells were isolated from the mice that received the adoptive transfer and stimulated with OVA. The proliferation of the CD4+ T cells from the two tissues was significantly inhibited in the Treg group compared with that in the control groups (Fig 4D). In addition, the excessive production of IL-4 and IFN-γ observed in the control groups was significantly reduced in the Treg group (Fig 4E). Together, these results indicate that adoptive transfer of the Tregs suppressed excessive activation of CD4+ T cells in the recipient EW-fed Rag23−3 mice, preventing the mice from developing food-allergic enteropathy. Thus, we conclude that Tregs derived from untreated Rag23−3 mice retain sufficient regulatory activity to suppress intestinal allergic responses.

**CD44$^{lo}$CD62L$^{hi}$CD4+ T cells (Naïve-like) in the mLN, but not in the spleen, of EW-fed Rag23−3 mice have the potential to differentiate into Tregs**

We have shown that CD4+ T cells of CN-fed Rag23−3 mice have the potential to differentiate into Tregs (Fig 2) when they are stimulated and expanded with anti-CD3 and anti-CD28 mAbs, but it was unclear whether the naïve CD4+ T cells in EW-fed Rag23−3 mice with enteropathy also keep their ability to differentiate into Tregs. Therefore, we next examined if the naive CD4+ T cells purified from the spleens or mLNs cells isolated from EW-fed Rag23−3 mice can differentiate into Tregs when they were stimulated with anti-CD3 and anti-CD28 mAbs. To purify the cells, we sorted naïve cells (naïve; CD44$^{lo}$CD62L$^{hi}$CD4+) or EMTs (effector/effector memory; CD44$^{hi}$CD62L$^{lo}$CD4+) from the mLN and spleen cells (SPL) of Rag23−3 mice fed with the EW- or CN-diet for 7 days (Fig 5A). The cells were then stimulated with plate-bound anti-CD3 and anti-CD28 mAbs under Treg-polarization conditions. After incubation for 72 h, the Treg population within the CD4+ T cell population was examined by flow cytometry (Fig 5B and 5C). The frequency of Tregs within the CD4+ T cell population in the culture of mLN-derived naïve T cells from the EW-fed Rag23−3 mice (EW_naive) was significantly lower than that in the cells from the culture of mLN-derived naïve T cells from CN-fed mice (CN_naive), but was significantly higher than that under the Treg-polarization culture of EMTs from EW-fed mice (EW_EMT) (Fig 5C). In the spleen cells (SPL), the frequency of Tregs was comparable in the Treg-polarization cultures of naïve phenotype CD4+ T cells (EW_naive) and EMTs (EW_EMT groups), but these were statistically lower than that in the culture of naïve T cells from CN-fed mice (Fig 5C). In addition, the production of the IL-4 was not enhanced in the naïve CD4+ T cells of the EW-fed mice (EW_naive) and was mostly comparable to that in the culture of naïve T cells of CN-fed mice (CN_naive); in contrast, overproduction of IL-4 in addition to IFN-γ was observed in the Treg-polarization culture of EMTs from EW-fed mice (EW_EMT). IL-2 production in mLN was comparable among experimental groups, but that in EW_EMT in SPL showed the highest level compared to those in other groups, but the difference of IL-2 production was not affected to the induction of Tregs (Fig 5D). These results indicate that naïve CD4+ T cells in the spleen and mLNs of Rag23−3 mice with severe allergic enteritis had a significantly reduced potential to differentiate into Tregs compared with those in mice fed the CN-diet fed the mice; this decrease of potential might be the result of changes of surface molecules expression of the cells affected by the allergic environment *in vivo* but not by the cytokines. This is consistent with the absence of detectable IL-4 or IFN-γ in the culture supernatant of Tregs differentiated from naive CD4+ T cells under polarization conditions, in contrast to the supernatant of Tregs differentiated from EMTs. Thus, this naïve CD4+ T cells might be one of several lineages of naïve T cells, not a typical naïve. Therefore, we defined the naïve CD4+ T cells defines as naïve-like T cells, hereafter.

In previous studies on Treg stability during tolerance acquisition in individuals with food allergy, CD137 expression by Tregs has been suggested to be a marker of enhanced Foxp3 expression and stability [14–16]. To examine the relationship between CD137 and Treg differentiation from naïve-like CD4+ T cells (EW_naive) and EMT phenotype (EW_EMT) from EW-fed mice, we examined the CD137 population within Tregs and CD44$^{hi}$CD62L$^{lo}$Treg populations differentiated from naïve-like T cells from CN- or EW-fed Rag23−3 mice (CN_naive, EW_naive) or EMTs of EW-fed Rag23−3 mice

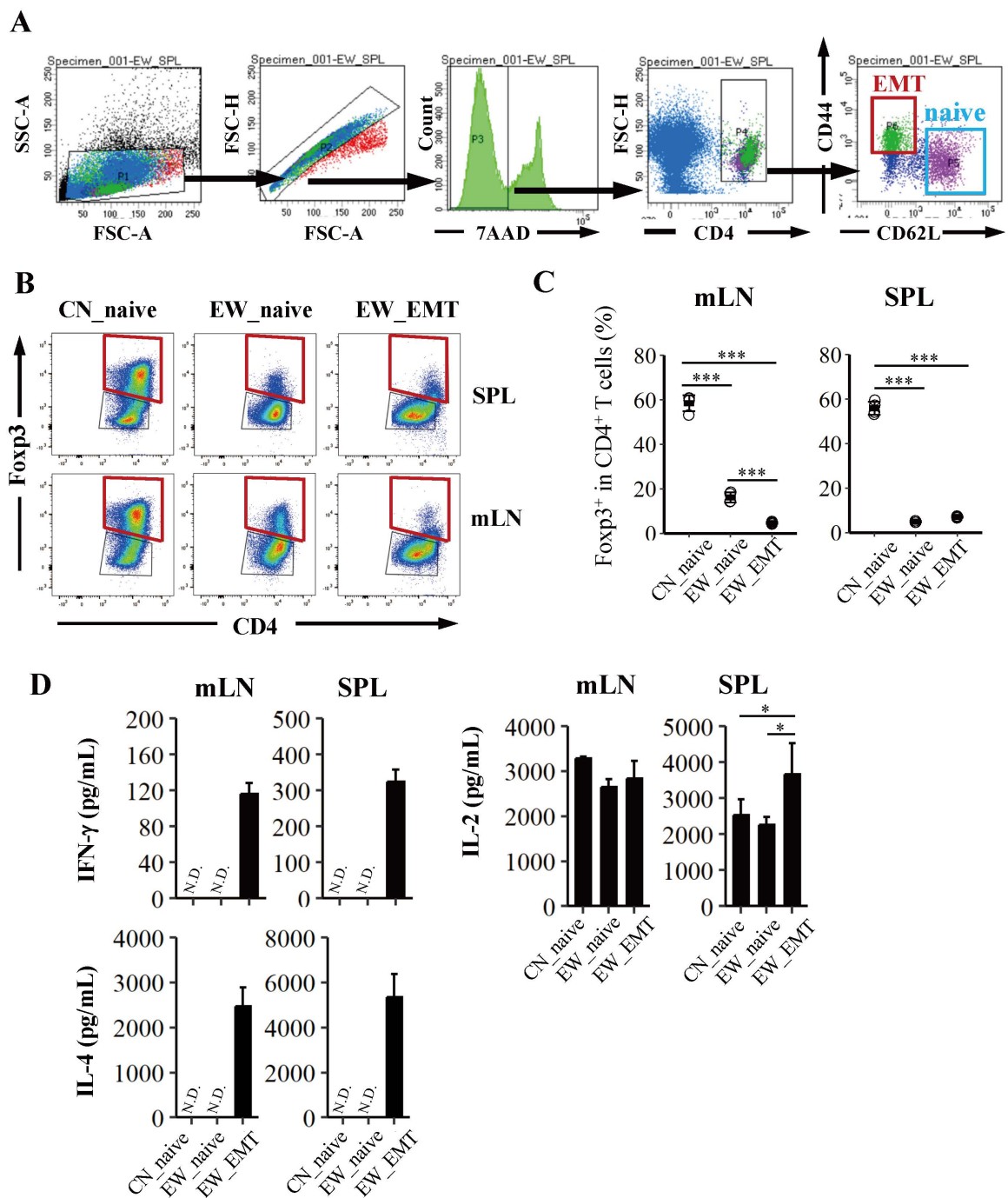

**Fig 5. Naïve-like CD4 $^+$ T cells (CD44$^{lo}$CD62L$^{hi}$) from mesenteric lymph nodes (mLN) of EW-fed Rag23−3 mice differentiate into regulatory T cells (Treg) even under severe allergic conditions.** A) Gating strategy to sort naïve (CD44$^{lo}$CD62L$^{hi}$) and effector/effector memory (EMT; CD44$^{hi}$CD62L$^{lo}$) CD4$^+$ T cells. Naïve CD4$^+$ T cells were prepared from Rag23−3 mice fed a diet containing casein (CN) or egg-white (EW), whereas EMTs were prepared only from Rag23−3 mice fed the EW-diet. B, C) Flow cytometry analysis of Treg population (Foxp3$^+$) in CD4$^+$ T cells differentiated from the naïve CD4$^+$ T cell (naïve) or EMT populations stimulated with plate-bound anti-CD3 and anti-CD28 antibodies and cultured under Treg polarization conditions for 72 h. Each plot indicates the value for an individual well, and the horizontal lines indicate the mean values (CN_naive: n=4, EW_naive: n=4, EW_EMT: n=4 using a mixture of cells from 6 [CN] or 3 [EW] mice/group). D), Interleukin (IL)-4 IFN-gamma (IFN-γ), and IL-2 levels in the culture supernatants (CN_naive: n=4, EW_naive: n=4, EW_EMT: n=4 using a mixture of cells from 6 [CN] or 3 [EW] mice/group). Data are representative of two independent experiments. Statistical analysis: Tukey's HSD test [*$p<0.05$, **$p<0.01$, and ***$p<0.001$ (between different groups)]. N.D., not detected.

(EW_EMT) (S5A and S5B Fig). The frequency of CD137+ cells within Tregs or CD44hiCD62LloTregs was significantly higher in the EW_naive and EW-EMT groups than those in the CN_naive group for both mLN and SPL, with almost 100% of the cells in the EW_EMT group being CD137+ cells (S5B Fig). In addition, the CD137+ population increased with increasing frequency of CD44hiCD62Llo within the Tregs population (S5C Fig), indicating that as the population of activated Tregs increased, so did the CD137+ population. However, the contribution of CD137 to the stability of the CD137+ population remains to be elucidated.

## Discussion

In this study, we examined whether allergen-specific T cells from food-allergic model mice showing weight loss and enteropathy are able to differentiate into functional Tregs with optimal activation and a stable suppressive function *in vitro*. To investigate this aim, we used two strains of OVA-specific T-cell-receptor transgenic mice: Rag23−3 mice (OVA23−3 mice crossed with *Rag2* knockout mice) and RagD10 mice (DO11.10 mice crossed with *Rag2* knockout mice). When fed a diet containing EW, RagD10 mice acquire tolerance to OVA, whereas Rag23−3 mice manifest severe Th2 responses and food-allergic enteropathy. Overall, our findings suggest that to obtain allergen-specific Tregs with sufficient activity to suppress food-allergic inflammation, naïve (CD44loCD62hi) CD4+ T cells must be from the total lymphocyte population, be expanded by stimulation with anti-CD3 and anti-CD28 mAbs but not OVA plus antigen-presenting cells, and then incubated under a cytokine milieu with low concentrations of IL-4 and IFN-γ. Furthermore, our results indicate that different from mLN, naïve T cells from the spleen of EW-fed Rag23−3 mice manifesting severe allergic responses, but not those from CN-fed Rag23−3 mice, had lost their ability to differentiate into Tregs before they were subjected to Treg expansion *in vitro*. This indicates not only that the ability of naïve CD4+ T cells to differentiate into Tregs is dependent on the conditions in the tissues in which they were localized *in vivo*, but also that the intestinal immune system maintains conditions suitable for Treg expansion even under conditions of severe allergic inflammation [30].

The only lymphocytes that Rag23−3 and RagD10 mice produce in response to OVA challenge are OVA-specific T cells. Using these mouse models allowed us to compare the functions of the OVA-specific Tregs between the inflammatory and tolerant conditions in response to exposure to the same volume of OVA over the same period of time. This comparison revealed that signaling via OVA-specific T-cell receptors plays a critical role in controlling Treg differentiation and inducing immune responses under conditions that cause T cells to respond excessively to allergens. More specifically, we have found that there are some differences in how the immune responses are invoked between the Rag23−3 and RagD10 mice. One difference, reported elsewhere, is that the two strains possess different T-cell-receptor genes [26]. Another difference involves the signals that induce IL-4 production by CD4+ T cells when the cells are stimulated with OVA through T-cell receptor. The signals to produce IL-4 are likely stronger and more immediate in OVA-specific EMTs from EW-fed Rag23−3 mice than in those from RagD10 mice, especially in the mLNs. Indeed, as we have described in a previous report [22], although CD4+ T cells from the spleens of EW-fed RagD10 mice were able to produce high levels of IL-4 and show a Th2-type immune response, it was increased only in the first 3 days of EW-feeding and was decreased thereafter out to 7 days; furthermore, CD4+ T cells from these mice stimulated with OVA needed 72 h to release high levels of IL-4 into the culture supernatant [22]. We have also reported that the Treg function induced during that experimental period is stable, as evidenced by the mice not showing severe allergic inflammation after restarting the EW diet after being on the CN diet for 1 month [22]. In contrast, high IL-4 production in mLN cells was immediately induced and maintained for 7–9 days in EW-fed R23-3 mice, and the incubation time for CD4+ T cells stimulated with OVA to produce a large amount of IL-4 was less than 48 h [22]. Inhibitory function of Tregs was not stable as indicated in the recurrence of the intestinal inflammation. Excessive IL-4 responses in the mLNs of EW-fed Rag23−3 mice dramatically decreased Treg induction and caused intestinal inflammation [22]. However, the mLNs are suggested to not only be a major inflammatory tissue in food allergy but also an important regulatory tissue in food allergy [22,31]; therefore, their potential to induce Tregs for regulation of intestinal inflammation may be higher than that of other tissues.

Although in the present study we found that T cells from EW-fed Rag23−3 mice have lower ability to differentiate into Tregs than T cells from RagD10 mice, we also found that using a combination of anti-CD3 and anti-CD28 mAbs to obtain sufficient volume of Tregs for adoptive transfer to other mice was superior to using a combination of antigen (OVA) and antigen-presenting cells. Although we did not investigate in detail why the responses to the Ab-stimulation were different from those to the antigen stimulation, similar findings, albeit with a different capacity of Tregs, have been reported elsewhere: Zhao et al. reported that the inhibitory effects on Th1 responses, surface molecules, and levels of Foxp3 expression in hen-egg-lysozyme–specific Tregs induced by mAbs are not the same as those induced by a combination of antigen plus antigen-presenting cells [24]. The implication of this result is that when considering adoptive Treg transfer as a therapeutic option, we need to choose the most appropriate approach to obtain Tregs with sufficient suppressive function. Our present results suggest that using mAbs to stimulate Treg differentiation affords Tregs with superior suppressive function in the context of severe food allergy. In addition, if we were to use the combination of antigen and antigen-presenting cells, we would also have to deal with removing the antigen and antigen-presenting cells from the culture at the end of the process to increase the purity of Tregs and avoid any potentially harmful effects of leftover antigen. Thus, the present findings highlight the potential of using Tregs differentiated from intestinal naïve T cells using a combination of anti-CD3 and anti-CD28 mAbs to suppress severe food-allergic inflammation.

Further studies are needed to elucidate the mechanism underlying the induction of suppressive Tregs by using mAbs. In the present study, induction of Tregs using the Abs-combination produced highly stable and highly suppressive Tregs, even from splenic CD4$^+$ T cells from untreated Rag23–3 mice, suggesting that activating naive T cells from untreated mice via both CD3 and CD28 mAbs can produce Tregs (iTregs) having persistent suppressive abilities in food allergy. However, in lung allergy, it has been reported that iTregs appear to enhance allergic responses [32]. Other reports have indicated that iTregs are stable, when they are expanded without the CD28 costimulatory signal [33]. Stimulation by anti-CD3 suppresses activation of EMTs induced by stimulation with co-stimulatory molecules such as anti-CD28 and Treg induction is promoted under Treg-polarization conditions [34]. Indeed, administration of anti-CD3 mAbs is already used clinically as a method to attenuate EMT activities for the treatment of some autoimmune diseases such as type 1 diabetes and multiple sclerosis [35–37]. Therefore, we hypothesize that if our iTregs are expanded via anti-CD3 stimulation only, they may show an increased suppressive capability.

The present results are noteworthy in that they show that, in EW-fed Rag23−3 mice with severe allergic inflammation, mLN had higher capacity to induce Tregs than spleen, and the frequency of Foxp3$^+$ Tregs in the CD4$^+$ T cell population induced from naïve-like (CD44$^{lo}$CD62$^{hi}$) CD4$^+$ T cells from mLN was higher than that from EMT (CD44$^{hi}$CD62$^{lo}$) CD4$^+$ T cells when the cells were cultured under Treg-polarization conditions and stimulated with anti-CD3 and anti-CD28 mAbs (Fig 5). This understanding will be useful for developing techniques to investigate the differences of Treg differentiation among different tissues in individuals with food allergy. However, the frequency of Tregs within the CD4$^+$ T cell population was significantly lower in the Tregs induced from naïve-like (CD44$^{lo}$CD62$^{hi}$) CD4$^+$ T cells from EW-fed Rag23−3 mice than in those from CN-fed the mice, and the Tregs from both naïve-like CD4$^+$ T cells from both types of mouse did not produce any cytokines, indicating that prior exposure of naïve CD4$^+$ T cells to excessive Th2-cytokine circumstances (i.e., severe allergy) may change the characteristics of the cells to naïve-like CD4$^+$ T cells, resulting in inhibition of Treg expansion in EW-fed mice compared with that in CN-fed mice. In previous studies examining Treg stability under allergic conditions [16–18], CD137 expression in Tregs has been suggested to be a marker of enhanced Foxp3 expression. In our mouse model of food-allergy enteropathy, although CD137 expression indicates activated Tregs, it may not necessarily reflect the correlation between the enhanced Foxp3 expression and stability of Treg function. Our data show the frequency of CD137$^+$ cells in CD44$^{hi}$CD62L$^{lo}$Foxp3$^+$CD4$^+$ T cells (activated Tregs) was enhanced by the EW diet, with 100% of the Tregs with the EMT phenotype being differentiated from EMTs from EW-fed Rag23−3 mice which expressing CD137 molecule (S5 Fig). This result reminds us of previous studies that have shown that activated Tregs have a higher plasticity and can change into EMTs [11,16–18]. Thus, our results suggest that naïve-like (CD44$^{lo}$CD62$^{hi}$) CD4$^+$ T cells expressing

a surface marker to migrate to the intestinal tissues (e.g., the gut homing integrin, α4β7 [38] or C-C motif chemokine receptor 9 (CCR9), specific homing receptors for colon or small intestine [39]) may be better for producing stable Tregs. In addition, ROR-γt Tregs also may be one of the functional population [20,40], but the characteristics of the Tregs from individuals with severe food allergies should be carefully investigated.

Regarding cooperative inhibition of Treg-differentiation by IL-4 and IFN-γ,it has been previously reported that IL-4 inhibits Treg induction [27], but the role of IFN-γ, which is produced by Th1 cells, is yet to be clarified. Although several reports have indicated that IFN-γ suppresses the induction of Tregs [41–43], IFN-γ is also reported to be required for the generation of Tregs in diseases, such as graft-versus-host disease or experimental autoimmune encephalomyelitis [44,45]. Although Th1 cells suppress the induction of Th2 cells, transplanting excess Th1 cells may not be an effective treatment method because of their ability to inhibit Treg differentiation (Fig 3). Because our results show that CD4+ T cells from a mouse model of allergy could still differentiate into Tregs under appropriate conditions of limited IL-4 and IFN-γ production, we consider that regulation of the cytokines may be useful for designing treatments using iTregs.

To clarify the impacts of the two cytokines, we performed the experiments adding rIL-4 and/or rIFN-γ or anti-IL-4 or IFN-γ mAbs into the Treg polarization culture; IFN-γ and IL-4 concentrations of the culture, the contribution of these cytokines to the inhibitory function of Treg differentiation and induction of EMTs were evaluated. We found that IFN-γ, IL-4, and EMTs worked together to create a feedback loop that works to balance the effects of the two cytokines; on the start of feeding Rag23−3 mice with EW, a greater amount of IL-4 might be strongly produced by EMT [22], excessive IL-4 production upregulated sensitivity to IFN-γ while also initiating EMT expansion, which results in IFN-γ overproduction of the EMT, which in turn promotes IL-4R expression, additional IL-4 overproduction, and further EMT expansion. Our results clearly indicated that during these inflammatory loops, Treg differentiation is strongly inhibited. Therefore, by the neutralization with each antibody in Fig 3B, the concentration of either cytokine was decreased, resulting that EMT activation was subsided, leading to recovery of Treg differentiation. In RagD10 mice, excessive IL-4 production can be induced, but it is produced more slowly and only for a short time during the early period of EW-feeding compared with what occurs in Rag23−3 mice [22] thereby preventing its inhibitory effects on Treg differentiation.

We also examined the expression of IL-4R and IFN-γR in Tregs differentiated from CD4+ T cells in RagD10 mice. We did not find any double-positive Tregs in the culture, but that might be the result of the low sensitivity of flow cytometry for small number of double-positive cells. Thus, these findings did not clarify whether the cytokines were produced by separate cells or there were cells expressing both cytokines. Adding anti IL-4 or IFN-γ Ab to the supernatant reduced the expression of both receptors compared with the situation without either of the Abs, and the frequency of Tregs in both groups was enhanced. That is, decreased cytokine production inhibited the receptor expression, showing that decreasing cytokine-mediated signaling improved the differentiation of Tregs. By adding of either IL-4 or IFN-γ or both, we clarified that CD4+ T cells from RagD10 mice are more sensitive to the IL-4–mediated than to the IFN-γ–mediated signal not to differentiate into Tregs.

It has been reported that excessive IL-4 production by naïve CD4+ T cells from the spleen of BALB/c mice induces IL-4R expression and phosphorylation of STAT6 and STAT3; the IL-4R expression is supplied by STAT3, and STAT3 suppresses IFN-γR1 through STAT6 [45] indicating that the receptor-mediated inter-regulatory relationship between the two transcription factors is provoked by excessive IL-4 and contribute to create the excessive IL-4 loop. Furthermore, IL-4 stimulation has been shown to downregulate CD28 and IFN-γR expression in CD3+CD4+CD8−IL-4R+ T cells [46]. Although it is unclear whether the relationship can be established in our Tregs differentiation, the addition of anti-IL-4 Ab to the culture may increase CD28 expression, increase the inhibitory effect of anti-CD3 Ab that suppresses T cell activation, and increase Treg differentiation [35]. Although we cannot discuss the IFN-γ-mediated mechanism underlying Treg expansion and their expression of IL-4R, there is possibly an inter-relationship of transcription factor–induced IFN-γ overproduction like excess production of IL-4 involved in the differentiation of our Tregs.

Adoptive transfer of Tregs for the treatment of severe inflammatory diseases is an attractive approach because it has the potential to not only treat but also prevent severe inflammatory diseases including allergy [3], although there have not yet been any clinical trials of Treg transfer therapy. There are, however, currently three approaches that have been reported to obtain stable Tregs from peripheral blood mononuclear cells: naturally occurring regulatory T cells (nTregs) [47], chimeric antigen receptor regulatory T cells (CAR-Tregs) [48], and stable and functional induced regulatory T cells (S/F-iTregs) [49]. It has been clarified in individuals who have outgrown their food allergy naturally or by OIT that Tregs and mediation of their function through inhibitory cytokines play important roles [3]. Therefore, Treg transfer may be a suitable approach for the treatment of individuals with persistent food allergy.

Allergen-specific Tregs are considered a good target not only for monitoring of the immunological condition of individuals with severe food allergy but also as a platform for the development of safe and stable treatments for these patients. The present results suggest several areas where further research is needed before Treg transfer can be used in the clinic. However, although we need to analyze the effect of stability of Treg function using the BALB/c mouse model [50] and verify the generality of this method, if we improve our understanding of T cell phenotypes and are able to find a specific marker of the cells related to the intestinal immune system, using antibodies rather than allergens and a low cytokine milieu for Treg expansion may provide a safe and efficient means of treating inflammatory diseases.

## Conclusions

Under severe allergic circumstances, induction of allergen-specific Tregs plays an important role in the inhibition of allergic inflammation, but there are many challenges to overcome before we will be able to obtain stable, suppressive Tregs for clinical use. Naïve ($CD44^{low}CD62L^{hi}$) $CD4^+$ T cells attributed to the intestinal immune system are candidate cells for expansion into stable, suppressive Tregs, but only when they are stimulated with anti-CD3 and anti-CD28 mAbs and rather than allergens and expanded under Treg polarization culture conditions with a low cytokine milieu.

## Supporting information

**S1 Fig. Body weight changes and jejunum histology in RagD10 and Rag23−3 mice fed a diet containing egg white (EW) or casein (CN).** A) RagD10 and Rag23−3 mice were fed a diet containing EW or CN for 7 days, and body weights were measured on days 0, 2, 4, 5, and 7. Body weight changes were determined relative to those on day 0. Values are expressed as mean ± SD. B) Jejunum histology. Samples of jejunum were collected from the mice on day 7 and stained with hematoxylin and eosin. Scale bars, 100 μm. n = 3 per group. Data are representative of two independent experiments. Statistical analysis: Tukey's HSD test [*$p < 0.05$ (Rag23−3 EW vs each of the three groups)].
(TIFF)

**S2 Fig. Sensitivity of $CD4^+$ T cells from RagD10 and Rag23−3 mice to stimulation by transforming growth factor beta 1 (TGF-β1).** Spleen (SPL) and mesenteric lymph nodes (mLN) were harvested from untreated RagD10 or Rag23−3 mice, and $CD4^+$ T cells were isolated by magnetic cell separation system. The cells were then stimulated with plate-bound anti-CD3 and anti-CD28 monoclonal antibodies and cultured in the presence of the indicated amounts of TGF-β1, retinoic acid (1 μM), and recombinant IL-2 (2 ng/mL) for 48 h, and the frequency of regulatory T cells ($Foxp3^+$) within the $CD4^+$ T cell population was determined. Each circle indicates the value for an individual well, and the horizontal lines indicate mean values (n = 3, mixture of cells from three mice/group). Data are representative of two independent experiments. Statistical analysis: Tukey's HSD test [(#$p < 0.05$ (between different strains); * $p < 0.05$, ** $p < 0.01$, and *** $p < 0.001$ (between different groups in each strain)].
(TIFF)

**S3 Fig. Cytokine profiles for the supernatants of $CD4^+$ T cells isolated from the spleen of RagD10 mice and Rag23−3 mice and cultured under regulatory T cell (Treg) polarization conditions.** RagD10 and Rag23−3 mice were

fed a diet containing egg white (EW) or casein (CN; control) for 7 days and spleens were harvested; CD4$^+$ T cells were isolated, stimulated with plate-bound anti-CD3 and anti-CD28 monoclonal antibodies, and cultured under Treg polarization conditions for 48 h, and the culture supernatants were collected. The concentrations of interleukin (IL)-2, IL-4, and interferon gamma (IFN-γ) in the supernatants were determined by enzyme-linked immunosorbent assay (n = 3, mixture of cells from three mice/group). Data are representative of two independent experiments. Statistical analysis: Tukey's HSD test [$^\#p < 0.05$ (CN vs EW in each strain); * $p < 0.05$, ** $p < 0.01$, and *** $p < 0.001$ (between different groups)]. N.D. = not detected.
(TIFF)

**S4 Fig. Expression of interferon-gamma (IFN-γ) or interleukin (IL)-4 receptor on regulatory T cells differentiated from untreated RagD10 mice is regulated by excessive IFN-γ or IL-4.** A) Frequency of CD25$^+$ cells in regulatory T cells (Tregs; Foxp3$^+$CD4$^+$ T cells) differentiated from splenocytes of untreated-RagD10 mice cultured under Treg-polarization culture conditions described in the caption to Fig 3B. B) Gating strategy for identifying IFN-γ receptor (IFN-γR$^+$) or IL-4 receptor (IL-4R$^+$) on CD4$^+$ T cells (left) and the frequencies of each receptor expression cells in CD4$^+$ T cells (right). Each plot indicates the value for an individual well and horizontal lines indicate mean values (n = 3, mixture of cells from three mice/group). C) Frequency of CD25$^+$ cells in Tregs from spleen cells of untreated RagD10 mice cultured under the Treg-polarization culture conditions indicated in Fig 3D and 3E. D) Frequencies of IFN-γR$^+$ or IL-4R$^+$ on CD4$^+$ cells supplemented with rIL-4 (left: 0, 0.5, 1.5, 2.5 ng/mL) or rIFN-γ (right: 0, 1.0, 2.0, 3.5 ng/mL). Error bars indicate means ± SD (n = 4, mixture of cells from two to three mice/group). Analysis: Tukey's HSD test {$^\#p < 0.05$ [IFN-γ (0 ng/mL)] vs [IFN-γ (3.5 ng/mL)] or [IL-4 (0 ng/mL)] vs [IL-4 (2.5 ng/mL)] in Fig S4C and S4D; * $p < 0.05$, ** $p < 0.01$, and *** $p < 0.001$ (between different groups)}.
(TIFF)

**S5 Fig. Effector/effector memory T cells within the regulatory T cell population show enhanced CD137 expression.** Naïve-like CD4$^+$ T cells (CD44$^{lo}$CD62L$^{hi}$) and effector/effector memory CD4$^+$ T cells (EMT; CD44$^{hi}$CD62L$^{lo}$) from Rag23−3 mice fed a diet containing egg white (EW) or casein (CN; control) for 7 days were stimulated with plate-bound anti-CD3 and anti-CD28 monoclonal antibodies and cultured under Treg-polarization culture conditions for 72 h. A) Gating strategy used to identify the CD137$^+$ subpopulation within the Foxp3$^+$ CD4$^+$ and CD44$^{hi}$CD62L$^{lo}$Foxp3$^+$CD4$^+$ populations. B) Frequency of CD137$^+$ cells within the Foxp3$^+$CD4$^+$ and CD44$^{hi}$CD62L$^{lo}$Foxp3$^+$CD4$^+$ T cell populations. C) Frequency of CD44$^{hi}$CD62L$^{lo}$ cells within the Foxp3$^+$CD4$^+$ T cell population. In panels 5B and 5C, plots indicate the values for individual wells and horizontal lines indicate mean values (CN_naive: n = 4, EW_naive: n = 4, EW_EMT: n = 4, using a mixture of cells from 6 (CN) or 3 (EW) mice/group). Data are representative of two independent experiments. Statistical analysis: Tukey's HSD test [*$p < 0.05$, ** $p < 0.01$, and *** $p < 0.001$ (between different groups)].
(TIFF)

## Author contributions

**Conceptualization:** Kyoko Shibahara, Haruyo Nakajima-Adachi.

**Data curation:** Kyoko Shibahara, Tomohiro Hoshino, Haruka Nakanishi, Haruyo Nakajima-Adachi.

**Formal analysis:** Kyoko Shibahara, Tomohiro Hoshino, Haruka Nakanishi, Kosuke Nishitsuji, Haruyo Nakajima-Adachi.

**Funding acquisition:** Satoshi Hachimura, Haruyo Nakajima-Adachi.

**Investigation:** Kyoko Shibahara, Tomohiro Hoshino, Haruka Nakanishi, Kosuke Nishitsuji, Yoshiyo Bamba, Haruyo Nakajima-Adachi.

**Methodology:** Kyoko Shibahara, Tomohiro Hoshino, Haruyo Nakajima-Adachi.

**Project administration:** Haruyo Nakajima-Adachi.

**Resources:** Haruyo Nakajima-Adachi.

**Supervision:** Haruyo Nakajima-Adachi.

**Validation:** Haruyo Nakajima-Adachi.

**Visualization:** Kyoko Shibahara, Tomohiro Hoshino, Haruka Nakanishi, Haruyo Nakajima-Adachi.

**Writing – original draft:** Kyoko Shibahara, Satoshi Hachimura, Haruyo Nakajima-Adachi.

**Writing – review & editing:** Tomohiro Hoshino, Haruka Nakanishi, Kosuke Nishitsuji, Kohei Soga, Satoshi Hachimura, Haruyo Nakajima-Adachi.

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
