## [Decision Letter · Decision Letter 0]

24 Jun 2024

PONE-D-24-17797Naïve intestinal T cells in food-allergic model mice, expanded with anti-CD3 and anti-CD28 antibodies while under regulatory T cell-polarization culture condition, can exhibit suppressive activityPLOS ONE

Dear Dr. Nakajima-Adachi,

Thank you for submitting your manuscript to PLOS ONE. After careful consideration, we feel that it has merit but does not fully meet PLOS ONE’s publication criteria as it currently stands. Therefore, we invite you to submit a revised version of the manuscript that addresses the points raised during the review process.

We look forward to receiving your revised manuscript.

Kind regards,

Masanori A. Murayama

Academic Editor

PLOS ONE

 [This work was supported by grants from the Kieikai Research Foundation (HNA, Grant number; 2017S063, https://www.nakashima-foundation.org/kieikai/) and 

Grant-in-Aid for Scientific Research (B) from

the Japan Society for the Promotion of Science (JSPS)(SH, Grant number; 26292065, https://www.jsps.go.jp/j-grantsinaid/index.html) .

The analysis for Fig. 5 and S5 Fig was funded by Meiji Holdings Co., Ltd (HNA and SH). ].  

[I have read the journal's policy and the authors of this manuscript have the following competing interests: [Meiji Holdings Co.,Ltd.]. 

Additional Editor Comments:

Thank you for submitting manuscript. This research is very interesting, however, this manuscript is open to discussion in this time.

As reviewer indicated, title is too difficult to understand the contents of this manuscript. To be honest, this manuscript is hard to understand in whole. So this manuscript need to help with English language editing.

As major point, this study just focused on the difference of R23-3 and RagD10, and the experiments and discussion was lack for the development of clinical application. Thus, this manuscript requires significant revision.

Reviewers' comments:

Reviewer's Responses to Questions

**Comments to the Author**

1. Is the manuscript technically sound, and do the data support the conclusions?

Reviewer #1: Partly

Reviewer #2: Yes

Reviewer #3: No

2. Has the statistical analysis been performed appropriately and rigorously? 

Reviewer #1: I Don't Know

Reviewer #2: Yes

Reviewer #3: Yes

3. Have the authors made all data underlying the findings in their manuscript fully available?

Reviewer #1: No

Reviewer #2: Yes

Reviewer #3: Yes

4. Is the manuscript presented in an intelligible fashion and written in standard English?

Reviewer #1: Yes

Reviewer #2: Yes

Reviewer #3: No

5. Review Comments to the Author

Reviewer #1: General comments

This study tried to find the difference in T cell characteristics between food allergic and non-allergic animal models. The article includes several interesting findings although there are some points which are requested to be improved to publish in PLOS ONE.

Specific comments

1. The title is difficult to understand the aim of this study.

2. I do not understand the reason why the authors stimulated T cells by anti-CD3 and anti-CD28 antibodies. What does the stimulation with the antibodies physiologically mean?

3. Related to above comment, I do not understand the reason why the responses to anti-CD3 and anti-CD28 were different from that to antigen stimulation.

4. The authors mentioned the Tregs from the mice as ‘allergen-specific Tregs”. I was wondering if the Tregs inhibited allergy in an allergen-specific manner. The authors should confirm the allergen specificity of the suppressive activity.

5. The authors wrote “ Naïve CD4+ T cells isolated from CN-fed Rag23-3 mice and stimulated with anti-CD3 and anti-CD28 mAbs differentiate into Tregs” (line 369). However, the authors used not only CD-fed mice but also EW-fed mice in the section. I do not understand the reason why title of the section was different from the content.

6. The position of markers in the graph right bottom of Fig. 2a seems out of alignment.

7. The authors wrote “Fig. 2c (data not shown).” (line 434). The authors should provide all data you used in this manuscript.

8. Fig. 3b shows that each of the anti-IL-4 or anti-IFN-g antibody almost completely inhibited the activity of the Sup by itself. I was wondering why IFN-g in anti-IL-4 condition or IL-4 in anti-IFN-g condition did not affect Treg induction.

9. I was wondering why the characteristics of naïve T cells of EW-fed mice differed from those of CN-fed mice in Fig. 5c. Were the naïve T cells of EW-fed mice really naïve?

10. It was very difficult to understand the purpose of each experiment throughout the manuscript. Please give it a clear title easy to understand the purpose of each section of results and explain the reason why the experiments were adopted for the purpose of each section.

Reviewer #2: <major comment="">

This manuscript showed a potential of ex-vivo polarization of functional Treg cells from the naïve intestinal T cells of on-going allergic enteropathy model mouse. Furthermore, overexpression of IL-4 and IFN-� from activated T cells suppressed the induction of Treg cells in the allergic mouse model. The experiments were conducted appropriately, and the detail of methods and results are clearly shown.

Although the length of manuscript is not limited in PONE journal, the text in method, result and figure legends is overlapping, that makes the redundant impression to the readers. Especially, the text in the figure legends may be shorter, because they use almost an equal volume as the text.

<minor points="">

1. Reference

The Ref. 1,2,4 should be updated to represent the recent clinical situations regarding food allergy and oral immunotherapy.

2. L. 371-373

Immobilized anti-CD3 stimulation should be TCR-mediated, although not antigen-specific stimulation.

3. L. 434

Fig 2c might be Fig 2b?

4. Statement of S3 Fig is missing in the text.

5. L.439-442

Blocking either IL-4 or IFN-� alone almost completely restored the suppressive effect of the Rag23-3_EW_Sup, suggesting these two cytokines are affecting to one lymphocyte simultaneously. How do authors explain this mechanism? Does a naïve T cell express both IL-4 receptor and IFN-� receptor on the surface? I’m wondering if direct supplementation of IL-4 and/or IFN-� shows the same effect?

6. L. 658-660

In this TCR-transgenic mouse model, how authors consider the difference between OVA+APC stimulation and anti-CD3+anti-CD28 stimulation? The difference of TCR signal transduction, or the other signals through co-stimulatory molecules on the APC? In other words, does OVA+APC stimulation with Treg-polarization condition on EW-fed Rag23-3 cells have a potential to induce Tregs?</minor></major>

Reviewer #3: In their manuscript "Naïve intestinal T cells in food-allergic model mice, expanded with anti-CD3 and anti-CD28 antibodies while under regulatory T cell-polarization culture conditions, can exhibit suppressive activity," Haruyo Nakajima-Adachi et al. compared two strains of OVA-specific TCR transgenic mice and found the distinct induction of Tregs and effector memory T cells in food allergen-mediated enteropathy.

1. This manuscript is difficult to follow and understand the emphasized points in its current version. The manuscript could benefit from English language editing to enhance clarity and readability. I strongly suggest this is necessary in its current format.

2. The study begins with the introduction of potential therapy using Treg administration for food allergy patients. Is there any clinical perspective on that? To avoid confusion for the reader, the authors should cite adequate references for Treg therapy in food allergies. Compared to other inflammatory diseases, cellular interventions such as cell therapy for food allergies represent an ultimate approach. The importance of how this research can be substantiated is in question.

3. Two strains of OVA-specific TCR tg mice are used in the manuscript, and they published the differential function of Tregs between these mouse strains in a 2017 PLOS One paper. In that paper, the amount of IL-4 was shown to be a causative factor in the regulation of Treg function. In this manuscript, their aim is to understand the stability of Tregs, but the study lacks analysis of how the cells are “stabilized”, for instance, how epigenetic regulations differ in the two mice lines of iTregs. How about RORgt+ Treg which is important cell population regulating allergic disorder. Is this caused by intrinsic control based on the strength of TCR signaling? How are Tregs differently induced in the two lines? The authors can experimentally define this by measuring the strength of TCR signaling.

4. Counter gating should be shown in Figure 1a of effector memory T cells with adequate control.

5. Upon activation, CD4+ T cells transiently express FOXP3, especially in activated CD4+CD25– T cells, thus CD25 expression should be shown in the panel throughout the manuscript.

6. I don’t understand why IL-2 is significantly induced in the control of Rag23-3 mice. Fully discuss this.

7. TCR is specific for OVA; however, EMT cells are significantly induced in the mice without OVA. What is the underlying mechanism of these phenotypic discrepancies?

8. The author indicated the importance of IL-4 for the induction of Treg. Does IL-4 administration induce Tregs in Rag23-3 CN mice?

9. TGF-beta should be referred to as “TGF-beta1” throughout the manuscript.

10. In terms of Treg induction via non-TCR mediated stimulation, I do not understand why the authors stimulated with an anti-CD3 antibody, which mimics TCR stimulation. The authors should reconsider the fundamental concept of this study and rearrange the experimental design.

11. The authors stop explaining the details of the results obtained about the different induction rates of Tregs between spleen and MLN by citing the Mucosal Immunology paper from Traxinger BR. The authors should fully explain and conclude the results based on their data.

12. IL-4 and IFNg are produced at higher levels in the Rag23-3 mice, thus Th1 and Th2 cells may be induced in the Treg-induction experiments. The authors should show if IL-4 and IFNg suppresses or skews Th1/Th2 cells by analyzing intracellular FACS of Th1/2 in mice.

13. Line 403: “The finding that OVA did not fully induce Treg differentiation, but stimulation with antibodies induced Treg differentiation in CN-fed Rag23-3 mice suggests the presence of factors” – however, they mimic TCR signaling by anti-CD3 and anti-CD28. I don’t understand this part.

14. The authors indicate Tregs derived from Rag23-3 mice have sufficient regulatory activity; however, it is important to examine the functional differences in the suppressive roles of RagDO10 and Rag23-3 derived Tregs in vivo and in vitro. Cytokine productions are different?

15. Lines 552 to 557 are hard to understand. Why do naïve T cells ameliorate differentiation to Tregs? This part is complicate to the reader. Please reorganize this part.

6. PLOS authors have the option to publish the peer review history of their article (what does this mean? ). If published, this will include your full peer review and any attached files.

**Do you want your identity to be public for this peer review?** For information about this choice, including consent withdrawal, please see our Privacy Policy .

Reviewer #1: No

Reviewer #2: No

Reviewer #3: No

---

## [Author Response · Author response to Decision Letter 1]

9 Nov 2024

Responses to the Editor’s Comments

Comment

Please ensure that your manuscript meets PLOS ONE's style requirements, including those for file naming. The PLOS ONE style templates can be found at https://journals.plos.org/plosone/s/file?id=ba62/PLOSOne_formatting_sample_title_authors_affiliations.pdf and https://journals.plos.org/plosone/s/file?id=wjVg/PLOSOne_formatting_sample_main_body.pdf

Responses

As requested, we have formatted the manuscript in line with PLOS ONE’s style requirements.

Comment 2

To comply with PLOS ONE submissions requirements, in your Methods section, please provide additional information regarding the experiments involving animals and ensure you have included details on (1) methods of sacrifice, (2) methods of anesthesia and/or analgesia, and (3) efforts to alleviate suffering.

Responses

As requested, we have added the following information regarding the experiments involving animals in Methods section:

“When necessary for cellular and histological analysis, mice were euthanized by cervical dislocation by experts. During the adoptive transfer of Tregs, to facilitate the subsequent injection of cells into the caudal vein, a mouse was placed in a small box with only its tail sticking out through the hole in the box for a few minutes. The injection was performed without anesthetization as smoothly and quickly as possible by experts.” (lines 147–152 in the revised manuscript)

“Our experiments were approved without setting preemptive humane endpoints, because during the experimental period (7 days), the mice did not exhibit other clinical signs than weight loss.” (lines 325–327 in the revised manuscript)

Comments 3 and 4

No. 3

We note that the grant information you provided in the ‘Funding Information’ and ‘Financial Disclosure’ sections do not match. When you resubmit, please ensure that you provide the correct grant numbers for the awards you received for your study in the ‘Funding Information’ section.

Responses

We provided the correct grant numbers for the awards we received for our study in the “Funding Information” section in our corrected manuscript as follows:

“This work was supported by grants from the Kieikai Research Foundation (HNA, Grant number; 2017S063, https://www.nakashima-foundation.org/kieikai/) and Grant-in-Aid for Scientific Research (B) from the Japan Society for the Promotion of Science (JSPS) (SH, Grant number; 26292065, https://www.jsps.go.jp/j-grantsinaid/index.html). The analysis for Fig 2a (CD25), Fig 3b–3g, Fig 5, S4 Fig, and S5 Fig was funded by Meiji Holdings Co., Ltd (HNA and SH), but we have no grant number assigned to this program grant in accordance with the nature of the grant, because The University of Tokyo's Corporate Sponsored Research Programs are programs established to conduct research on common issues that are of highly public nature in collaboration with the University of Tokyo, using funds received from the private sector and other external organizations. [Corporate Sponsored Research Programs | The University of Tokyo (u-tokyo.ac.jp)]”

No. 4

Thank you for stating the following financial disclosure:

[This work was supported by grants from the Kieikai Research Foundation (HNA, Grant number; 2017S063, https://www.nakashima-foundation.org/kieikai/) and

Grant-in-Aid for Scientific Research (B) from

the Japan Society for the Promotion of Science (JSPS) (SH, Grant number; 26292065, https://www.jsps.go.jp/j-grantsinaid/index.html) .

The analysis for Fig. 5 and S5 Fig was funded by Meiji Holdings Co., Ltd (HNA and SH).

Responses

We have included the following Role of Funder statement in the cover letter to this response document:

“The funders had no role in the study design, data collection and analysis, decision to publish, or preparation of the manuscript.”

In addition, in this revision, we have added several data obtained by some experiments. These experiments have been performed by the funding support by Meiji Holdings Co., Ltd. Thus, we would like to update this financial disclosure as follows:

“This work was supported by grants from the Kieikai Research Foundation (HNA, Grant number; 2017S063, https://www.nakashima-foundation.org/kieikai/) and Grant-in-Aid for Scientific Research (B) from the Japan Society for the Promotion of Science (JSPS) (SH, Grant number; 26292065, https://www.jsps.go.jp/j-grantsinaid/index.html). The analysis for Fig 2a (CD25), Fig 3b–3g, Fig 5, S4 Fig, and S5 Fig was funded by Meiji Holdings Co., Ltd (HNA and SH), but we have no grant number assigned to this program grant in accordance with the nature of the grant, because The University of Tokyo's Corporate Sponsored Research Programs are programs established to conduct research on common issues that are of highly public nature in collaboration with the University of Tokyo, using funds received from the private sector and other external organizations. [Corporate Sponsored Research Programs | The University of Tokyo (u-tokyo.ac.jp)].

The funders had no role in the study design, data collection and analysis, decision to publish, or preparation of the manuscript.”

This update of financial disclosure is included in our cover letter.

Comment 5

Thank you for stating the following in the Competing Interests section:

[I have read the journal's policy and the authors of this manuscript have the following competing interests: [Meiji Holdings Co.,Ltd.].

Response

We have added the following text to the Competing Interests section of our revised manuscript:

“all of the authors of this manuscript had read the journal’s policy and that the authors had a competent interest with Meiji Holdings Co., Ltd. This does not alter our adherence to PLOS ONE policies on sharing data and materials.”

This statement is also declared in the cover letter to this response document.

Comment 6

We note that you have included the phrase “data not shown” in your manuscript. Unfortunately, this does not meet our data sharing requirements. PLOS does not permit references to inaccessible data. We require that authors provide all relevant data within the paper, Supporting Information files, or in an acceptable, public repository. Please add a citation to support this phrase or upload the data that corresponds with these findings to a stable repository (such as Figshare or Dryad) and provide and URLs, DOIs, or accession numbers that may be used to access these data. Or, if the data are not a core part of the research being presented in your study, we ask that you remove the phrase that refers to these data.

Response

As requested, we have removed the phrase “data not shown” from the revised manuscript. In addition, we have corrected the citation for Fig 2c to Fig 2b and now cite S3 Fig (corrected S4 Fig) as shown in line 496–497 in the revised manuscript as follows:

“The cytokine profiles of the supernatants used in this experiment are shown in S3 Fig, and these profiles were consistent with those shown in Fig 2b.”

Comment 7

Please include captions for your Supporting Information files at the end of your manuscript, and update any intext citations to match accordingly. Please see our Supporting Information guidelines for more information: http://journals.plos.org/plosone/s/supporting-information.

Response

As requested, we have added captions for our Supporting Information files (lines 1093–1158) at the end of our revised manuscript meeting PLOS ONE's style requirements.

Comment 8

Thank you for submitting manuscript. This research is very interesting, however, this manuscript is open to discussion in this time. As Reviewer indicated, title is too difficult to understand the contents of this manuscript. To be honest, this manuscript is hard to understand in whole. So, this manuscript needs to help with English language editing. As major point, this study just focused on the difference of R23-3 and RagD10, and the experiments and discussion were lack for the development of clinical application. Thus, this manuscript requires significant revision.

Response

We appreciate the opportunity to revise and resubmit our manuscript. We have updated the title to more simply describe our study to readers. In addition, the English in the revised manuscript has been edited by two native-English-speaking professional scientific editors from ELSS, Inc. (http://www.elss.co.jp), who also provided suggestions on ways to improve the clarity of the revised manuscript and according to the suggestion, the original manuscript have been throughout updated. In the Discussion, we now discuss the clinical application (line 889–899 in the revised manuscript). The remaining changes that we have made in response to the Reviewer’s comments are explained on the following pages. In addition, to describe the possibility of clinical application of treatment of Treg-transfer in the intestinal diseases, we added (ref. 15: Int J Mol Sci. 2020;21:7015).

Responses to Reviewer 1

Comment 1

The title is difficult to understand the aim of this study.

Response

We have updated the title as follows:

Old title: “Naïve intestinal T cells in food-allergic model mice, expanded with anti-CD3 and anti-CD28 antibodies while under regulatory T cell-polarization culture condition, can exhibit suppressive activity”

New title: “Ovalbumin-specific regulatory T cells with the naïve phenotype (CD62LloCD44hi) from mesenteric lymph nodes suppress food-allergic enteropathy in mice, when expanded by anti-CD3/CD28 antibodies but not by ovalbumin plus antigen-presenting cells”

Comment 2

I do not understand the reason why the authors stimulated T cells by anti-CD3 and anti-CD28 antibodies. What does the stimulation with the antibodies physiologically mean?

Response

First, we observed that T cells from Rag23-3 mice have lower ability to differentiate into Foxp3+ regulatory T cells (Tregs) than T cells from RagD10 mice. However, to promote T cell differentiation into Tregs, we needed an approach for stimulating and expanding the T cells. In addition, in performing an adoptive transfer of Tregs as a treatment, we have to differentiate and expand CD4+ T cells into Tregs and to generate “poly-clonal Tregs” in vitro after isolating T cells from the individuals. If we used an antigen for the expansion in the culture, we would have to remove both the antigen-presenting cells and the antigen from the culture to increase the purity of the Tregs and to prevent side effects caused by contamination with the antigen. In addition, it has been reported that the functions and phenotype of Tregs differ depending on whether their differentiation was induced by using an antigen plus antigen-presenting cells or by using a combination of anti-CD3 and anti-CD28 antibodies (ref. 24: Cellular Immunol: 2014;290:179–184). Thus, in the present study, we examined the stimulation of T cell differentiation into Tregs by using anti-CD3 and anti-CD28 antibodies as well as OVA plus antigen-presenting cells. We now describe this rationale and a comparison of the data obtained with both approaches (Fig 1: antigen/antigen-presenting cell approach; Fig 2: antibody approach) in the revised Introduction (line 113–119), Discussion (lines 769–788) and at the beginning of the Results (lines 422–427). See also our response to Comment 3.

Comment 3

Related to above comment, I do not understand the reason why the responses to anti-CD3 and anti-CD28 were different from that to antigen stimulation.

Response

As Reviewer 1 indicates, we did not clarify in our original manuscript the reasons for the different outcomes when Treg differentiation is induced by using the antigen/antigen-presenting cell approach or the antibody approach. As mentioned in our response to the previous comment, there is a report showing that there are functional and phenotypic differences between the iTreg lineages induced using either an antigen (HEL)/antigen-presenting cell approach or the combination of anti-CD3/anti-CD28 antibodies (ref. 24: Cellular Immunol. 2014;290:179–184). Therefore, our result has a precedent. However, considering together the present and previous comments from the Reviewer, we have added the following sentences to the revised Discussion to explain the different outcomes from the different approaches to induce Treg differentiation (lines 769–788); the reference mentioned in our responses to comments 2 and 3 has also been added to the revised manuscript as reference 24:

“Although in the present study we found that T cells from EW-fed Rag23-3 mice have lower ability to differentiate into Tregs than T cells from RagD10 mice, we also found that using a combination of anti-CD3 and anti-CD28 mAbs to obtain sufficient volume of Tregs for adoptive transfer to other mice was superior to using a combination of antigen (OVA) and antigen-presenting cells. Although we did not investigate in detail why the responses to the Ab-stimulation were different from those to the antigen stimulation, similar findings have been reported elsewhere: Zhao et al. reported that the inhibitory effects on Th1 responses, surface molecules, and levels of Foxp3 expression in hen-egg-lysozyme–specific Tregs induced by mAbs are not the same as those induced by a combination of antigen plus antigen-presenting cells [24]. The implication of this when considering adoptive Treg transfer as a therapeutic option is that we have to choose the most appropriate approach to obtain Tregs with sufficient suppressive function. Our present results show that using mAbs to stimulate Treg differentiation affords Tregs with superior suppressive function in the context of severe food allergy. In addition, if we were to use the combination of antigen and antigen-presenting cells, we would also have to deal with removing the antigen and antigen-presenting cells from the culture at the end of the process to increase the purity of Tregs and avoid any potentially harmful effects of leftover antigen. Thus, the present findings highlight the potential of using Tregs differentiated from intestinal naïve T cells using a combination of anti-CD3 and anti-CD28 mAbs to suppress severe food allergic inflammation.”

In addition, in the Introduction, we now clearly indicate the reason for using antibody-mediated stimulation rather than antigen-mediated stimulation as follows (lines 113–119 in the revised manuscript):

“It has been reported that the functions and phenotypes of Tregs differentiated by using a combination of anti-CD3 and anti-CD28 monoclonal antibodies (mAbs) differ from those of Tregs differentiated with an antigen in the presence of antigen-presenting cells [24]. Thus, in the present study, we examined how to obtain stable suppressive Tregs from the Rag23-3 food-allergic enteropathy model by comparing both the inducing means and their resultant differentiation levels and functions of Tregs induced in Rag23-3 and RagD10 mice when fed EW.”

Comment 4

The authors mentioned the Tregs from the mice as ‘allergen-specific Tregs”. I was wondering if the Tregs inhibited allergy in an allergen-specific manner. The authors should confirm the allergen specificity of the suppressive activity.

Response

Both the Rag23-3

---

## [Decision Letter · Decision Letter 1]

6 Jan 2025

PONE-D-24-17797R1Ovalbumin-specific regulatory T cells with the naïve phenotype (CD62LloCD44hi) from mesenteric lymph nodes suppress food-allergic enteropathy in mice, when expanded by anti-CD3/CD28 antibodies but not by ovalbumin plus antigen-presenting cellsPLOS ONE

Dear Dr. Nakajima-Adachi,

Thank you for submitting your manuscript to PLOS ONE. After careful consideration, we feel that it has merit but does not fully meet PLOS ONE’s publication criteria as it currently stands. Therefore, we invite you to submit a revised version of the manuscript that addresses the points raised during the review process.

We look forward to receiving your revised manuscript.

Kind regards,

Masanori A. Murayama, Ph.D.

Academic Editor

PLOS ONE

Additional Editor Comments:

Thank you for re-submission of your manuscript. This revised manuscript is interesting, but it has open issues. Please check comments from reviewer2. And I have some comments; In introduction section, please explain why authors use different OVA-TCR Tg mice. At least, authors should explain what is the difference OVA23-3 and DO11.10 mice at basic levels.

In Results section, authors should delete the supplementary explanation about fundamental knowledge, and some sentence should move discussion section. To promote understanding the contents, authors need concise result section. I think the result section is too long. As pointed out by reviewer 2, conclusions are unclear.

At all figures, it was very difficult to understand what is the significantly difference. Please use * or #, instead of a, b, c, and d, and use lines between graphs. And please change small to large alphabet, Fig1a -> Fig1A. And please change IFN-gamma -> IFNγ, TGF beta1 - TGFβ1.

In result section, authors investigated some cytokine from helper T cells, but not IL-10 and IL-17. Why authors did not investigate IL-10 expression?

Reviewers' comments:

Reviewer's Responses to Questions

**Comments to the Author**

1. If the authors have adequately addressed your comments raised in a previous round of review and you feel that this manuscript is now acceptable for publication, you may indicate that here to bypass the “Comments to the Author” section, enter your conflict of interest statement in the “Confidential to Editor” section, and submit your "Accept" recommendation.

Reviewer #1: All comments have been addressed

Reviewer #2: All comments have been addressed

Reviewer #3: (No Response)

2. Is the manuscript technically sound, and do the data support the conclusions?

Reviewer #1: Yes

Reviewer #2: Yes

Reviewer #3: Partly

3. Has the statistical analysis been performed appropriately and rigorously? 

Reviewer #1: (No Response)

Reviewer #2: Yes

Reviewer #3: Yes

4. Have the authors made all data underlying the findings in their manuscript fully available?

Reviewer #1: Yes

Reviewer #2: Yes

Reviewer #3: Yes

5. Is the manuscript presented in an intelligible fashion and written in standard English?

Reviewer #1: Yes

Reviewer #2: Yes

Reviewer #3: No

6. Review Comments to the Author

Reviewer #1: The authors appropriately answered to my questions and comments, and revised the manuscript according to the answers.

Reviewer #2: The manuscript has been appropriately edited, and the methods and results sound much clearer. The role of excessive amount of IL-4 and IFN-gamma, either autologous production or supplementation, on the suppression of Treg differentiation has now been clearly shown.

Reviewer #3: In their manuscript, "Naïve intestinal T cells in food-allergic model mice, expanded with anti-CD3 and anti-CD28 antibodies under regulatory T cell-polarization culture conditions, can exhibit suppressive activity," the authors currently describe, "Ovalbumin-specific regulatory T cells with the naïve phenotype (CD62L^lo^CD44^hi^) from mesenteric lymph nodes suppress food-allergic enteropathy in mice, when expanded by anti-CD3/CD28 antibodies but not by ovalbumin plus antigen-presenting cells." Haruyo Nakajima-Adachi et al. compared two strains of OVA-specific TCR transgenic mice and found distinct differences in the induction of regulatory T cells (Tregs) and effector memory T cells during food allergen-mediated enteropathy.

The comparison between anti-CD3/CD28 antibodies and ovalbumin plus antigen-presenting cells is a critical aspect of the paper. I suggest a revised title for the manuscript: "Stable and suppressive regulatory T cells from mesenteric lymph nodes suppress food-allergic enteropathy in mice" as the current title is long and somewhat complex.

Extracting Tregs from mesenteric lymph nodes (MLN) for therapeutic use in food allergy patients seems impractical. Would it not be possible to use Tregs derived from peripheral blood instead as an experimental setting which strength the requirement of their study? The manuscript appears to lack appropriate control experiments to address this question. The authors mention in line 79 that “it is difficult to separate allergen-specific Tregs from the total population of peripheral blood mononuclear cells.” However, obtaining Tregs from MLN seems even more challenging than isolation from peripheral blood mononuclear cells (PBMCs).

The authors highlight that iTregs derived from naïve CD4+ T cells of OVA-transgenic mice strongly suppress allergies. However, their explanation about how antigen-specific naïve T cell-derived Tregs could be obtained in non-transgenic models is overly complex for a general understanding of immunological mechanisms.

There are several established mouse models for food allergy, such as OVA-Alum. It would be more robust to test the expansion of Tregs using these models, comparing the two methods described in the manuscript. Demonstrating the inhibition of allergic symptoms with the authors’ method would strengthen the study’s findings.

Additionally, the manuscript cites a new reference (Reference 24). However, the cited paper’s results seem contradictory to the authors’ findings, stating that “iTregs generated by activation with antigen/APC are more suppressive than iTregs generated by anti-CD3/CD28 antibodies.” The cited manuscript attributes these functional differences to chemokine-related gene expression. If the authors choose to reference this paper, they should address this apparent discrepancy in their discussion.

Overall, the manuscript leaves its conclusions unclear. It does not effectively explain what was discovered regarding the differences between the two OVA-TCR mouse strains or the mechanisms of Treg induction by different antigen stimulation methods. While the stated aim is to evaluate the stability of Tregs, the study lacks critical analysis, such as how epigenetic regulation might contribute to stabilization in the two strains of iTregs. To provide generalizable insights, I recommend additional experimental comparisons of Tregs derived from these strains to clarify their cellular properties.

7. PLOS authors have the option to publish the peer review history of their article (what does this mean? ). If published, this will include your full peer review and any attached files.

**Do you want your identity to be public for this peer review?** For information about this choice, including consent withdrawal, please see our Privacy Policy .

Reviewer #1: No

Reviewer #2: No

Reviewer #3: No

---

## [Author Response · Author response to Decision Letter 2]

17 Apr 2025

Responses to the Editor’s Comments

Comment 1

In introduction section, please explain why authors use different OVA-TCR Tg mice. At least, authors should explain what is the difference OVA23-3 and DO11.10 mice at basic levels.

Responses

As requested, we have added the following information about the differences of basic immune responses in these OVA-TCR Tg mice in the Introduction section in the revised manuscripts:

“The mice predominantly produce IFN-γ by stimulation with anti-CD3 antibody or OVA, indicating Th1-biased responses [22].” (lines 103 – 105 in the revised manuscript)

“and strong IL-4 responses, when they were stimulated with anti-CD3 antibody or OVA [22].” (lines 107 – 108 in the revised manuscript).

In addition, to clarify our aim and the differences of the 2 strains of mice, we corrected some words in the Abstract section and highlighted by yellow.

Comment 2

In Results section, authors should delete the supplementary explanation about fundamental knowledge, some sentence should move discussion section.

Responses

As the Editor commented, it might be better to move some sentences to the Discussion section.

However, we felt that it is difficult to understand our results, especially Figure 2 and 3, without the supplementary explanation of the results and the subsequent discussion of the results, so we have added the explanation at the site after the description of the results of the figures, especially Figure 2 and 3. However, in the Results section of Figure 2, we agree that the description explaining the results was not appropriate. Therefore, we have corrected some sentences (lines 434 – 435, 441– 445 as highlighted by yellow.

Therefore, we would like to submit our revised manuscript without any changes regarding the supplementary explanation, although we can understand the editor's comment.

Comment 3

it was very difficult to understand what is the significantly difference. Please use * or #, instead of a, b, c, and d, and use lines between graphs. And please change small to large alphabet, Fig1a -> Fig1A. And please change IFN-gamma -> IFNγ, TGF beta1 - TGFβ1.

Responses

We have corrected the way to show the significances by different letters, such as a, b, c, and d to * or # in all figures and each legend (highlighted by yellow) and corrected small to large alphabet like from Fig1a to Fig 1A in the revised manuscript. In addition, we have corrected IFN-gamma to IFNγ, and TGF beta1 - TGFβ1.

Comment 4

In result section, authors investigated some cytokine from helper T cells, but not IL-10 and IL-17. Why authors did not investigate IL-10 expression?

Responses

As the editor suggested, we further analyzed IL-10 production in the supernatant of MLN or spleen CD4+ T cells incubated under in vitro Treg-polarization culture condition stimulated with anti-CD3 and anti-CD28 antibodies. The results were added in Fig. 2B. The results showed that Treg-population induced in Rag23-3 or RagD10 mice differed in IL-10 production when stimulated and differentiated with anti-CD3 and CD28 antibodies.

Therefore, we have corrected the legend of Figure 2 (line 491), have added the ELISA kit used in this assay in the Methods section (lines 243 – 244) and further added the following sentences in the Results section of our revised manuscript:

“The level of IL-10 production in the spleen cells and in the mLN cells was significantly higher in the Rag23-3 mice than in the RagD10 mice when fed the EW-diet. The amount of IL-10 produced by CD4+ T cells in mLN was much more than that in spleen EW-fed mice.” (lines 465 – 468 in the revised manuscript)

“the production of IL-4, IFN-γ, and IL-10 in EW-fed Rag23-3 mice. In addition, Tregs differentiated from CD4+ T cells by stimulation with mAbs in both strains of EW-fed mice probably have sufficient regulatory function producing IL-10, but suppressive function of the Tregs in both strains of CN-fed mice may be independent of IL-10.” (lines 470 – 473 in the revised manuscript)

Others

We apologize for the lack of description of the result describing IL-2 production in the supernatant in Fig 5D of our previous manuscript. Therefore, we have added the following sentences in the Results section (lines 685 – 688) of the revised manuscript as follows;

“IL-2 production in mLN was comparable among experimental groups, but that in EW_EMT in SPL showed the highest level compared to those in other groups, but the difference of IL-2 production was not affected to the induction of Tregs (Fig 5D).”

Responses to Reviewer 3

Comment1

I suggest a revised title for the manuscript: "Stable and suppressive regulatory T cells from mesenteric lymph nodes suppress food-allergic enteropathy in mice" as the current title is long and somewhat complex.

Responses

Following the suggestion of Reviewer3, we have changed the title of our revised manuscript.

Old title: "Ovalbumin-specific regulatory T cells with the naïve phenotype (CD62LloCD44hi) from mesenteric lymph nodes suppress food-allergic enteropathy in mice, when expanded by anti-CD3/CD28 antibodies but not by ovalbumin plus antigen-presenting cells".

We agree with the critical comments of Reviewer3 that our manuscript should reduce the generality but emphasize the specificity of the method inducing regulatory T cells from the mouse model of severe intestinal allergy. However, the unique way in which we obtained the suppressive and stable regulatory T cells in this study even under severe food allergic conditions, is that by using anti-CD3 and anti-CD28 antibodies for stimulation, naive antigen-specific CD4+ T cells which are likely to originate from the intestinal tissues, are differentiated into Tregs. Therefore, if Reviewer3 agrees with our proposal, we think it would be better to add the words "ovalbumin-specific regulatory T cells differentiated from naïve phenotype (CD44loCD62Lhi)”, “mesenteric lymph nodes”, and “severe food allergy” in the title. Furthermore, we would like to clearly state our claim in the title, as the title suggested by Reviewer3 is similar to our short title “Stable and suppressive regulatory T cell induction in a food-allergic enteropathy mouse model”.

We would appreciate it if Reviewer3 and the Editor could reconsider and accept the title we changed based on Reviewer 3's comments in the revised manuscript as follows:

New title: “Ovalbumin-specific regulatory T cells differentiated from the naïve phenotype (CD44loCD62Lhi) in mesenteric lymph nodes stably suppress enteropathy even in severe food-allergic mice "

Comment2

Extracting Tregs from mesenteric lymph nodes (MLN) for therapeutic use in food allergy patients seems impractical. Would it not be possible to use Tregs derived from peripheral blood instead as an experimental setting which strength the requirement of their study? The manuscript appears to lack appropriate control experiments to address this question. The authors mention in line 79 that “it is difficult to separate allergen-specific Tregs from the total population of peripheral blood mononuclear cells.” However, obtaining Tregs from MLN seems even more challenging than isolation from peripheral blood mononuclear cells (PBMCs).

Responses

As Reviewer3 suggested, we agree that extracting Tregs from MLN for therapeutic use would be impossible in food allergic patients. Therefore, we proposed the use of marker molecule (such as alpha4beta7) migrating to the intestine, expressed on naïve phenotype T cells to enable the method possible in clinical use; as described in lines 828 – 830 in the discussion section of our previous manuscript;

“Therefore, our results suggest that naïve-like (CD44loCD62hi) CD4+ T cells expressing a surface marker to migrate to the intestinal tissues (e.g., the gut homing integrin, alpha4beta7 [38]) may be best for producing stable Tregs”.

By considering Reviewer3’s comment, we thought that the description (line 78 – 79) in the Introduction section of the previous manuscript may give a misunderstanding to the readers of PLOS ONE, because allergic researchers have made efforts to find the maker of Tregs in peripheral blood. In addition, in the Discussion section, we mentioned about the marker migrating to the intestinal tissue of Tregs in the peripheral blood. Actually, although it was in graft-versus-host disease, C-C motif chemokine receptor 9 (CCR9), specific homing receptors for colon or small intestine, was presented as a candidate of transplantation Treg marker migrating to the intestine and reducing the symptom (American J Transplantation, 2023:23;1102-1115, https://doi.org/10.1016/j.ajt.2023.01.030), although they were obtained from spleen. However, they are trying to transplant Tregs migrating to the intestine and cure the disease. Therefore, we have corrected the sentence in the Introduction section and added CCR9 molecule as a candidate molecule and this paper as a reference in the Discussion section of the revised manuscript.

<Introduction section>

Old

*lines 78 – 81: however, it is difficult to separate allergen-specific Tregs from the total population of peripheral blood mononuclear cells, because of the lack of specific surface markers of Tregs with stable inhibitory functions that will not produce off-target effects, although CD137 has been proposed as a candidate marker [16-18].

New

*lines 78 – 80 in the revised manuscript: “many allergists are trying to find markers for stable Tregs specifically activated by antigens in peripheral blood, but the goal has not been achieved, although CD137 has been proposed as a candidate marker [16-18]”.

<Discussion section>

Old

*lines 828 – 830: Thus, our results suggest that naïve-like (CD44loCD62hi) CD4+ T cells expressing a surface marker to migrate to the intestinal tissues (e.g., the gut homing integrin, alpha4beta7 [38]) may be best for producing stable Tregs.

New

*lines 844 – 847 in the revised manuscript: “Thus, our results suggest that naïve-like (CD44loCD62hi) CD4+ T cells expressing a surface marker to migrate to the intestinal tissues (e.g., the gut homing integrin, α4β7 [38] or C-C motif chemokine receptor 9 specific homing receptors for colon or small intestine, CCR9 [39]) may be better for producing stable Tregs.”

*Reference:

39. Larson JH, Jin S, Loschi M, Bolivar Wagers S, Thangavelu G, Zaiken MC, et al. Enforced gut homing of murine regulatory T cells reduces early graft-versus-host disease severity. Am J Transplant. 2023;23:1102-1115. https:// doi: 10.1016/j.ajt.2023.01.030

Comment3

The authors highlight that iTregs derived from naïve CD4+ T cells of OVA-transgenic mice strongly suppress allergies. However, their explanation about how antigen-specific naïve T cell-derived Tregs could be obtained in non-transgenic models is overly complex for a general understanding of immunological mechanisms.

There are several established mouse models for food allergy, such as OVA-Alum. It would be more robust to test the expansion of Tregs using these models, comparing the two methods described in the manuscript. Demonstrating the inhibition of allergic symptoms with the authors’ method would strengthen the study’s findings.

Responses

We are grateful for Reviewer3’s suggestion to make our results more robust as a better way to induce functional Tregs. However, we have confirmed that the food-allergic inflammation and recovery from the severe inflammation observed in the EW-fed TCR-transgenic mouse model of OVA23-3, such as enteropathy, bone loss, tolerance acquisition, and mast cell infiltration into the intestinal tissues are reproduced by a food-allergic enteropathy model established by using BALB/c mice fed with EW-diet after intraperitoneal sensitization with OVA and alum. (Burggraf M., Nakajima-Adachi H., et al. Mol. Nutr. Food Res. 2011;55:1475–1483). In our manuscript, we have not yet analyzed in detail the characteristics of Tregs in the BALB/c mouse model and therefore, we agree to confirm the reproducibility of the induction of functional Tregs in the BALB/c mouse model. Indeed, we can analyze OVA-specific Treg responses by using the BALB/c mouse model (Morinaga H., Allergol Int. 2020;69:622-625). However, even if the method of inducing Tregs used in the present study worked in the BALB/c mouse model, we would raise the question whether it was the effect of alum immunity and whether the results might be different if we used the other adjuvant. Also, in allergic diseases, effective treatments vary from person to person and may not work for everyone. Therefore, we have added a sentence in the Discussion section suggesting this Treg induction method as one of them in severe allergy and added a reference in our revised manuscript as follows;

“although we need to analyze the effect of stability of Treg function using the BALB/c mouse model [50] and verify the generality of this method,” (lines 920 – 921 in the revised manuscript)

*Reference:

50. Burggraf M, Nakajima-Adachi H, Hachimura S, Ilchmann A, Pemberton AD, Kiyono H., et al. Mol. Nutr. Food Res. 2011;55:1475-1483. doi 10.1002/mnfr.201000634)

Comment4

Additionally, the manuscript cites a new reference (Reference 24). However, the cited paper’s results seem contradictory to the authors’ findings, stating that “iTregs generated by activation with antigen/APC are more suppressive than iTregs generated by anti-CD3/CD28 antibodies.” The cited manuscript attributes these functional differences to chemokine-related gene expression. If the authors choose to reference this paper, they should address this apparent discrepancy in their discussion.

Responses

Thank you for your suggestion. Although the study in reference 24 showed that the iTregs generated by anti-CD3 and anti-CD28 antibodies were less suppressive than iTregs generated by stimulation with HEL, which was the antigen of the Tregs, and APC, the situation and aim of the experiments were different from ours. We indicated in our previous manuscript that we could induce iTregs from severe allergic mice by using anti-CD3 and anti-CD28 antibodies, not antigen/APCs by the following mechanism that the mice were unable to induce Tregs when they were stimulated with antigen and APC, because of overproduction of IL-4 and IFN-γ. Therefore, we don’t think there is a discrepancy with reference24, but to avoid any misunderstanding of our aim, we have added some words in line 790 and corrected sentences in lines 794 – 796 in the revised manuscript, and both corrections are highlighted in yellow as follows;

“Although we did not investigate in detail why the responses to the Ab-stimulation were different from those to the antigen stimulation, similar findings, albeit with a different capacity of Tregs, have been reported elsewhere: Zhao et al. reported that the inhibitory effects on Th1 responses, surface molecules, and of Foxp3 expression levels in hen-egg-lysozyme–specific Tregs induced by mAbs are not the same as those induced by a combination of antigen plus antigen-presenting cells [24]. The implication of this result is that when considering adoptive Treg transfer as a therapeutic option, we need to choose the most appropriate approach to obtain Tregs with sufficient suppressive function. Our present results suggest that”　

Comment 5

Overall, the manuscript leaves its conclusions unclear. It does not effectively explain what was discovered regarding the differences between the two OVA-TCR mouse strains or the mechanisms of Treg induction by different antigen stimulation methods. While the stated aim is to evaluate the stability of Tregs, the study lacks critical analysis, such as how epigenetic regulation might contribute to stabilization in the two strains of iTregs. To provide generalizable insights, I recommend additional experimental comparisons of Tregs derived from these strains to clarify their cellular properties.

Responses

We are grateful for suggesting the importance comparing epigenetic status between Tregs from Rag23-3 as a severe food-allergic enteropathy mouse model and those from RagD10 as a tolerance acquisition mouse model. However, our study got started at the finding that in Rag23-3 mice, iTregs could not be induced by stimulation with OVA plus APCs. Therefore, it is the most important finding in

---

## [Editor Report · Decision Letter 2]

22 Apr 2025

Ovalbumin-specific regulatory T cells differentiated from the naïve phenotype (CD44loCD62Lhi) in mesenteric lymph nodes stably suppress enteropathy even in severe food-allergic mice

PONE-D-24-17797R2

Dear Dr. Haruyo Nakajima-Adachi,

We’re pleased to inform you that your manuscript has been judged scientifically suitable for publication and will be formally accepted for publication once it meets all outstanding technical requirements.

Kind regards,

Masanori A. Murayama, Ph.D.

Academic Editor

PLOS ONE

Additional Editor Comments (optional):

Thank you for submitting revised manuscript. I am pleased for sending this mail. This manuscript is completely revised according to reviewer's comments. Thus, my decision is accept in this revision. Congratulations.
---

## [Editor Report · Acceptance letter]

PONE-D-24-17797R2

PLOS ONE

Dear Dr. Nakajima-Adachi,

I'm pleased to inform you that your manuscript has been deemed suitable for publication in PLOS ONE. Congratulations! Your manuscript is now being handed over to our production team.

Kind regards,

on behalf of

Dr. Masanori A. Murayama

Academic Editor

PLOS ONE